# CauKer: Classification Time Series Foundation Models Can Be Pretrained on Synthetic Data

**Shifeng Xie**
Université Paris Cité
Huawei Noah's Ark Lab
xidianxieshifeng@gmail.com

**Vasilii Feofanov**
Huawei Noah's Ark Lab
42.com

**Jianfeng Zhang**
Huawei Noah's Ark Lab

**Themis Palpanas**
Université Paris Cité

**Ievgen Redko**
Huawei Noah's Ark Lab

## ABSTRACT

Time series foundation models (TSFMs) have recently gained significant attention due to their strong zero-shot capabilities and widespread real-world applications. Such models typically require a computationally costly pre-training on large-scale, carefully curated collections of real-world sequences. To allow for a sample-efficient pre-training of TSFMs, we propose CauKer, a novel algorithm designed to generate diverse, causally coherent synthetic time series with realistic trends, seasonality, and nonlinear interactions. CauKer combines Gaussian Process (GP) kernel composition with Structural Causal Models (SCM) to produce data for sample-efficient pre-training of state-of-the-art classification TSFMs having different architectures and following different pre-training approaches. Additionally, our experiments reveal that CauKer-generated datasets exhibit clear scaling laws for both dataset size (10K to 10M samples) and model capacity (1M to 783M parameters), unlike real-world datasets, which display irregular scaling behavior. The source code is publicly available at https://github.com/ShifengXIE/CauKer.

## 1 INTRODUCTION

Time series data are ubiquitous in applications ranging from healthcare (Gnassounou et al., 2025) and human activity recognition (Chen et al., 2025a) to industrial monitoring (Susto et al., 2018). Recently, the time series community has devoted significant effort to developing large-scale pre-trained time series foundation models (TSFMs). Inspired by advances in natural language processing and computer vision, these models aim to achieve strong zero-shot performance in out-of-distribution (OOD) settings. TSFMs have been proposed for both forecasting (Ansari et al., 2024; Woo et al., 2024; Bhethanabhotla et al., 2024) and classification tasks (Goswami et al., 2024; Lin et al., 2024; Feofanov et al., 2025), showing promising results. TSFMs are usually trained on large-scale pre-training dataset collections gathered from different application domains. Recent works used as many as 300 billion timepoints for model pre-training (Shi et al., 2025).

Despite the prevalence of large-scale pre-training in the development of TSFMs, several works (Hoo et al., 2024; Dooley et al., 2023; Taga et al., 2025; Liu et al., 2025) showed that comparable performance can be achieved by training them purely on synthetic data. The latter approach has several important advantages. First, it removes the need for time-consuming data collection and curation. This is especially important in time series classification that lacks diverse and rich pre-training corpora. Second, it allows for generating arbitrarily large datasets for model scaling. Finally, it makes the OOD evaluation more meaningful, mitigating the risk of data leakage. Inspired by the recent success of foundation models in tabular classification (Hollmann et al., 2023), our paper proposes a novel sample-efficient pre-training framework for TSFMs in classification based purely on synthetic data. Contrary to tabular and forecasting synthetic data generation pipelines, our proposal seeks to generate sequences with meaningful correlations between samples and realistic temporal

dependencies within them. We provide an in-depth, large-scale study of its benefits compared to pre-training on commonly used time series classification corpora.

**Findings**   Overall, our findings can be summarized as follows:

1. A carefully designed synthetic data generation pipeline can be efficiently used in training classification TSFMs. We propose such a pipeline and show that it requires rethinking synthetic data generators proposed previously for tabular data and time series forecasting.

2. Pre-training on synthetic data reveals clear scaling laws both in terms of dataset size and model size. We illustrate this finding by showing that such scaling laws are broken when using common classification benchmarks for pre-training, likely due to the lack of diversity in existing classification datasets.

3. Distinct from forecasting (Yao et al., 2025), where the leaderboard (with the exception of (Hollmann et al., 2023)) is still dominated by models pre-trained on large-scale real-world datasets, we show that pre-training on solely synthetic data can lead to state-of-the-art performance in classification.

The rest of this paper is organized as follows. In Section 2, we present recent advances in TSFMs and describe commonly used pre-training datasets. In Section 3, we present the problem setup considered in our work and the proposed synthetic data generation pipeline. In Section 4, we empirically validate the effectiveness of CAUKER-generated synthetic data through extensive experiments, demonstrating its strong generalization, scalability, and superiority over existing synthetic generation methods. Finally, we conclude our work and its limitations in Section 5.

## 2   RELATED WORK

**Time series foundation models**   Recent advances in TSFM have followed two primary directions: (1) training models from scratch on large-scale, diverse time series datasets (Ansari et al., 2024; Goswami et al., 2024; Das et al., 2024; Gao et al., 2024; Rasul et al., 2024; Wang et al., 2024; Woo et al., 2024; Bhethanabhotla et al., 2024; Lan et al., 2025; Gao et al., 2024; Lin et al., 2024; Liu et al., 2024b; Cohen et al., 2024; Auer et al., 2025), and (2) leveraging large language models (LLMs) as backbones for time series tasks (Chang et al., 2023; Gruver et al., 2024; Zhou et al., 2023; Xue & Salim, 2023; Cao et al., 2023; Jin et al., 2023; Liu et al., 2024a). The first approach focuses on developing architectures specifically tailored for time series, while the second approach explores encoding time series data into textual formats or extending the model's input mechanisms to natively handle sequential numeric data. Among the TSFMs mentioned above, a vast majority were proposed for time series forecasting, with only (Feofanov et al., 2025; Gao et al., 2024; Goswami et al., 2024; Chang et al., 2025; Lin et al., 2024; Zhang et al., 2025) natively supporting time series classification. In particular, (Feofanov et al., 2025; Lin et al., 2024; Roschmann et al., 2025) specifically target classification by contrastively pre-training encoder-only models over time series gathered from popular classification benchmarks. They achieve state-of-the-art results in this task. Goswami et al. (2024) is an encoder-decoder model used for classification and other popular time series tasks, such as forecasting, imputation, and anomaly detection. Gao et al. (2024) relies on a custom architecture and is used in generative and prediction tasks by leveraging task-specific tokens. Finally, Chang et al. (2025) fine-tunes an LLM by adding an appropriate encoder for input data and a classification head to generate predictions.

**Pre-training datasets**   The training data for TSFM generally fall into three categories: real-world, synthetic, or hybrid datasets combining the two. Models trained (or fine-tuned in case of LLM-based TSFMs) exclusively on real data (Das et al., 2024; Gao et al., 2024; Rasul et al., 2024; Wang et al., 2024; Feofanov et al., 2025; Gao et al., 2024; Lin et al., 2024; Chang et al., 2023; Gruver et al., 2024; Zhou et al., 2023; Xue & Salim, 2023; Cao et al., 2023; Jin et al., 2023) typically leverage extensive collections (ranging from 300k to 50M distinct time series) drawn from diverse domains such as traffic, finance and environmental monitoring. Training on these datasets, however, may be suboptimal scaling-wise as Quan et al. (2024) obtained comparable performance using $< 1\%$ of the original 27B pre-training dataset from (Woo et al., 2024), while Yao et al. (2025) showed that famous forecasting TSFMs have very flat scaling laws in the multivariate setting. Meanwhile, forecasting

models such as Chronos (Ansari et al., 2024) and TimesFM (Das et al., 2024) enhance their training corpus by incorporating synthetic time series data alongside real-world data. Beyond sequence-native TSFMs, there is a complementary line of work (Chen et al., 2025b) that maps time series into image-like representations and then applies vision Transformers. Finally, such methods as TimePFN (Taga et al., 2025) and ForecastPFN (Dooley et al., 2023) are pre-trained solely on synthetic data. In all these forecasting models, synthetic data is commonly generated through structured statistical procedures, including Gaussian process (kernel-based) methods or piecewise linear and seasonal pattern constructions with additive noise (for more details, we refer the interested reader to Appendix A.) To the best of our knowledge, no prior work has proposed classification-oriented synthetic data generation methods for training time series foundation models.

## 3 OUR CONTRIBUTIONS

We now introduce the task of zero-shot time series classification using TSFMs. We then formally present the common pre-training strategies and introduce our synthetic data generation pipeline.

### 3.1 PROBLEM SETUP

**Zero-shot classification** As done in prior work on unsupervised representation learning (Franceschi et al., 2019; Yue et al., 2022), we see a TSFM as an encoder $F : \mathbb{R}^t \to \mathbb{R}^q$ that is kept frozen during the evaluation. For a downstream classification dataset $\mathcal{D} = \{(x_i, y_i)\}_{i=1}^n$ with labels $y_i \in \{1, \ldots, C\}$, we use a TSFM to obtain embeddings $z_i = F(x_i)$ and train a lightweight classifier $h : \mathbb{R}^q \to \{1, \ldots, C\}$ solely on $\{(z_i, y_i)\}$. At test time, an unseen series $x^*$ is classified by $\hat{y} = h\big(F(x^*)\big)$. As $F$ is kept frozen, the resulting accuracy measures the quality of its learned representations.

To quantify OOD generalization ability, we follow Yao et al. (2025) and evaluate the studied TSFMs only on samples not seen during their pre-training. In practice, if we evaluate a given TSFM on a test set from a UCR (Dau et al., 2019) dataset, we ensure that the TSFM was not pre-trained on it. Our CAUKER-pretrained models are trained only on CAUKER-generated synthetic series and never see UCR (or any real-world classification benchmark) during pre-training. The original Mantis and MOMENT (Feofanov et al., 2025; Goswami et al., 2024) checkpoints, as well as other TSFMs we compare to, are pre-trained on large real-world corpora that include UCR train sets (but never UCR test data), following the standard protocol in prior work. Therefore, original Mantis and MOMENT are, to some extent, in the in-distribution setup.

**Self-supervised pre-training** Self-supervised learning (SSL) has emerged as a powerful training paradigm for foundation models, allowing them to effectively learn discriminative representations from large-scale unlabeled datasets, significantly reducing dependency on costly data labeling (Jaiswal et al., 2020). SSL methods are categorized into two principal types: contrastive learning and masked (reconstruction) learning (Liu et al., 2023). Contrastive learning focuses on distinguishing between similar (positive) and dissimilar (negative) data pairs to learn meaningful representations. Conversely, masked learning leverages reconstruction objectives by training models to predict masked parts of the input, thereby gaining robust contextual understanding (Zhang et al., 2022).

In our work, we cover both pre-training regimes. To this end, we consider Mantis (Feofanov et al., 2025), an open-source FM pre-trained contrastively, and MOMENT (Goswami et al., 2024), which is a masked-based pre-trained model. Detailed formulations of the loss functions and architecture specifics for these models are provided in the Appendix B.

### 3.2 CAUKER: SYNTHETIC DATA GENERATION FOR TIME SERIES CLASSIFICATION

We now present our proposed synthetic data generation pipeline, termed CAUKER for **Cau**sal-**Ker**nel generation. To develop our intuition about it, we note that the synthetic data for the time series classification task needs to combine two key ingredients. On the one hand, the generated sequences should exhibit common time series patterns such as seasonality, periodicity, and trend. On the other hand, successful classification assumes that individual time series have a meaningful clustering structure that allows the trained model to successfully learn how to disentangle the underlying clusters during training. Below, we present a generation pipeline that satisfies these desiderata.

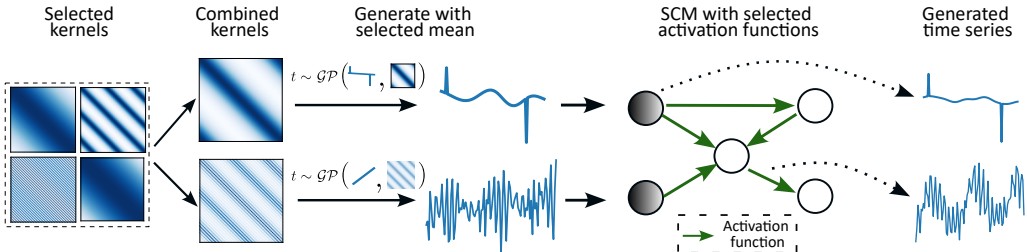

Figure 1: An illustration of the proposed CAUKER pipeline. Kernels sampled from the kernel bank $\mathcal{K}$ are randomly combined and used together with sampled mean functions to form GP priors. Time series sampled from these GP priors act as root nodes in a directed acyclic graph that encodes causal dependencies between nodes. Each edge of this graph applies an activation function from a predefined activation function bank and aggregates over incoming edges using a random linear transformation to propagate transformed time series through the graph. Intermediate node outputs are optionally interpolated to fixed length, forming the final synthetic dataset. This procedure yields rich, diverse, and causally consistent time series for self-supervised pre-training.

**Proposed approach** To proceed, we now define three banks of functions, namely: kernel, mean and activation banks denoted as $\mathcal{K} = \{\kappa_i(t,t')\}_{i=1}^{n_\mathcal{K}}$, $\mathcal{M} = \{\mu_i(t)\}_{i=1}^{n_\mathcal{M}}$ and $\mathcal{A} = \{\sigma(t)_i\}_{i=1}^{n_\mathcal{A}}$, respectively. For the kernel bank, we use the same kernel functions as Ansari et al. (2024). For mean functions, we consider a linear function $ax + b$, exponential function $ae^{bx}$, and anomaly mean function that inserts random values from $\mathcal{U}(-5,5)$ at random indexes. Finally, the activation functions we use for $\mathcal{A}$ are a linear function $ax + b$ with $a \sim \mathcal{U}(0.5,2)$, $b \sim \mathcal{U}(-1,1)]$, ReLU activation, sigmoid, sine function, element-wise modulo operation $x \bmod c$ for $c \sim \mathcal{U}[1,5]$, and Leaky ReLU with a random negative slope from $\mathcal{U}(0.01,0.3)$. For simplicity, in what follows we let $\{s_i\}_{i=1}^n \sim \mathcal{S}$ denote an i.i.d. sampling (without replacement) of $n$ elements from a set $\mathcal{S}$.

Our generative pipeline, illustrated in Figure 1, then proceeds in five steps as follows:

- Step 1. **Kernel bank sampling** We start by sampling candidate kernels from the kernel bank, *ie*, $\{\kappa_i(t,t')\}_{i=1}^K \overset{\text{i.i.d.}}{\sim} \mathcal{K}$ for some random number of candidate kernels $K \sim \mathcal{U}(1, n_\mathcal{K})$.

- Step 2. **Kernel composition** We define a composite kernel based on $K-1$ randomly sampled binary operations (+ and ×). More formally, for a random sequence $\{\star_i\}_{i=1}^{K-1} \sim \{+, \times\}$, we let $\kappa^* = \kappa_1(t,t') \star_i \cdots \star_{K-1} \kappa_K(t,t')$.

- Step 3. **Root nodes generation** We draw $M$ mean functions $\{\mu_i(t)\}_{i=1}^M \overset{\text{i.i.d.}}{\sim} \mathcal{M}$, $M \sim \mathcal{U}(1, n_\mathcal{M})$ and repeat Step 1 and Step 2 $M$ times to obtain composite kernels $\{\kappa_i^*\}_{i=1}^M$. We further define $M$ GP priors to sample from $\{\mathcal{GP}(\mu_i, \kappa_i^*)\}_{i=1}^M$.

- Step 4. **Activation bank sampling** We sample a set of $E$ activation functions from the activation bank, *ie*, $\{\sigma_i\}_{i=1}^E \sim \mathcal{A}$, $E \sim \mathcal{U}(1, n_\mathcal{A})$.

- Step 5. **Causal graph propagation** We randomly generate a directed acyclic graph (DAG) $(\mathcal{V}, \mathcal{E})$ with $|\mathcal{E}| = E$, $|\mathcal{V}| = V$, and $M < V$ *root nodes*, i.e., nodes with in-degree zero. We then define a bijection $\phi : \mathcal{V} \rightarrow \{\sigma_1, \sigma_2, \ldots, \sigma_V\}$ such that each node $v_i$ is uniquely associated with a function $\sigma_l$, i.e., $\phi(v_i) = \sigma_l$. We then associate a time series $t_i \in \mathbb{R}^L$ sampled from $\mathcal{GP}(\mu_i, \kappa_i^*)\}$ to each of the $M$ root nodes. The value $t_{v_j}$ associated with a given non-root vertex $v_j$ is then calculated as follows. First, we concatenate all incoming edges $e_{\cdot j}$ and aggregate them using a randomly initialized linear layer with weights and biases $W, b \sim \mathcal{N}(0,1)$, then we apply a randomly sampled activation function to get $t_{v_j} = \phi(v_j)(W \times [e_{\cdot j}] + b)$.

A complete pseudocode of this procedure, as well as the composition and visualizations of the kernel, mean, and activation banks, are provided in Appendix C.

**Design choices** The synthetic datasets generated using our CAUKER approach effectively encode diverse, realistic patterns and causal dynamics characteristic of real-world classification problems. Unlike the kernel-only generator of Ansari et al. (2024) (Steps 1,2), which was designed for forecasting

and therefore draws zero-mean Gaussian-process samples that emphasize smooth trend extrapolation, our task calls for retaining the mean level itself (Step 3) as a discriminative cue – a choice that is empirically confirmed in Section 4.4. Conversely, the structural causal model (SCM) generator (Steps 4,5) originally proposed for tabular classification (Hollmann et al., 2023) produces rich non-linear dependencies but lacks hallmark time series motifs such as seasonality or linear trends. By unifying kernel composition with an SCM processing, CAUKER inherits the periodic structure of Gaussian processes while simultaneously injecting causal semantics through directed edges. Finally, we note that different nodes of the same SCM in CAUKER can be interpreted as different channels of a multivariate time series that share a common causal structure. This hints at the potential of CAUKER for pre-training inherently multivariate models as well. We further ablate the structure of the SCM in C.3 and the impact of kernel bank composition in C.2.

**Positioning our method**    While synthetic data generation has been explored for time series forecasting and tabular classification, adapting these pipelines to time series classification TSFMs is far from trivial. Forecasting-oriented generators (e.g., kernel-based method with zero means(Ansari et al., 2024)) are optimized for smooth extrapolation and often neglect class-conditional structure and inter-class separability. Conversely, tabular SCM generators (e.g., TabPFN(Hollmann et al., 2023)) discard temporal structure. Our method, CAUKER, bridges this gap by unifying kernel-composed Gaussian processes with structural causal models, resulting in synthetic corpora that exhibit discriminative clusters and well-behaved scaling laws, in sharp contrast to current real-world classification collections whose heterogeneous, imbalanced composition makes them unreliable for pre-training of TSFMs.

Our experiments in Section 4 demonstrate that foundation models pre-trained on such data exhibit improved out-of-distribution generalization and meaningful scaling behavior, outperforming models trained solely on traditional synthetic benchmarks and performing with those trained on much larger real-world time series corpora. We restrict our pre-training experiments to univariate inputs and treat each node's trajectory as an individual series, in order to match the univariate pre-training regime of TSFMs.

## 4    EXPERIMENTAL RESULTS

We now empirically evaluate the effectiveness of our proposed CAUKER framework for pre-training classification TSFMs. Our experiments aim to answer the following key questions:

**Q1**. How does CAUKER compare to alternative synthetic data generation methods?

**Q2**. Do TSFMs trained on CAUKER data exhibit meaningful data and model scaling laws?

**Q3**. Can CAUKER-generated synthetic data be a competitive replacement for real-world benchmarks in training TSFMs?

In all our experiments, we consider two recent TSFMs, namely Mantis and MOMENT. Mantis is an 8M encoder-only model pre-trained using contrastive learning. We use the 77M version of the MOMENT model. The latter is an encoder-decoder model pre-trained based on masked reconstruction. Considering these two models allows us to compare two different pre-training paradigms as previously done in (Yao et al., 2025) for forecasting. Finally, we follow Feofanov et al. (2025) and evaluate Mantis in a zero-shot regime by learning a Random Forest classifier on the embeddings of training examples of a given dataset. For MOMENT, Goswami et al. (2024) evaluated their model using an Support Vector Machine classifier. For both models, we report the test accuracy averaged over 128 UCR datasets, where each dataset has train and test sets following (Dau et al., 2019).

### 4.1    Q1: CAUKER AGAINST ALTERNATIVE SYNTHETIC GENERATORS

**Experimental setup**    To better understand the exact contribution of the proposed CAUKER, we first start by establishing the virtues of our synthetic data generation pipeline compared to prior work. For this, we generate four different synthetic corpora, namely: 1) FPFN (Taga et al., 2025) that uses a linear model of coregionalization to sample multivariate time series, 2) KernelSynth (Ansari et al., 2024) that randomly composes covariance kernels to define a Gaussian process with zero mean; 3) Mean+KernelSynth: our re-implementation of the KernelSynth baseline in which we

additionally add non-zero mean functions in the GP; 4) SCM (TabPFN generator), a reconstruction of the structural-causal model proposed by Hollmann et al. (2023) for tabular classification [1]. We generate univariate time series with length $T = 512$ as both Mantis and MOMENT were trained on time series of this length. For a fair comparison, we fix the number of synthetic samples to 100K.

Table 1: Average zero-shot accuracy (%) on the UCR benchmark after pre-training on synthetic corpora generated by different methods.

| Model | SCM | FPFN | KernelSynth | Mean-KernelSynth | CAUKER (ours) |
|---|---|---|---|---|---|
| Mantis | 73.49 | 77.52 | 77.70 | 78.20 | **78.31** |
| MOMENT | 59.23 | 70.85 | 69.31 | 72.56 | **74.24** |

**Results** Table 1 shows a relative comparison of our proposal compared to other methods. Our first observation is that classification-tailored tabular data generation pipeline SCM underperforms significantly compared to all other methods. This suggests that temporal dependencies are important for time series classification, differently from the forecasting setup, where TabPFN trained using SCM-generated data is among the strongest foundation models. We further note that forecasting-tailored FPFN and Kernel-Synth also provide suboptimal results, even more so for MOMENT. In the case of Mantis, the results of pre-training on these two datasets are closer to the reported performance of the Mantis model. This can be likely explained by the architecture of Mantis that incorporates strong time series classification priors into it (mean, standard deviation, and difference encoding in the token generator unit). On the contrary, MOMENT is a generic encoder-decoder model. We further note a distinct positive effect of including non-mean functions in the GP used to generate time series in our pipeline. Finally, CAUKER improves upon this stronger baseline in both cases, highlighting the additional benefit of causal structure. The last two observations are particularly valid for MOMENT, indicating that they compensate for the lack of useful inductive biases for the task of time series classification.

**Computational cost.** To quantify the computational overhead introduced by the SCM layer, we compared CAUKER to the KernelSynth generator that uses the same composite kernel bank and GP implementation as our method. Both generators were asked to produce $N = 1,000$ univariate time series of length $T = 512$ under identical hardware and software settings. As reported in Table 2, CAUKER is in fact slightly faster than the KernelSynth baseline. Table 2 further decomposes the CAUKER runtime: more than 99% of the time attributed to the generator itself is spent in GP kernel sampling, while constructing the SCM graph and propagating signals through it contributes less than 1% of the total cost.

Table 2: Overall wall-clock generation time and internal runtime breakdown for CAUKER.

| Item | Time (s) |
|---|---|
| CAUKER | 121.64 |
| KernelSynth | 182.25 |
| GP kernel sampling | 118.54 |
| SCM structure + propagation | 1.14 |

This happens because CAUKER samples GPs only for the root nodes of the SCM; a single set of root processes is then propagated through the causal graph, from which multiple nodes can be extracted as different univariate series.

**Qualitative analysis** We now provide insights for as why CAUKER is particularly suitable for classification. Intuitively, we expect that having a discriminative signal in the generated data – a clustering structure defining meaningful groups of time series – should enable efficient classification on previously unseen samples. To verify this, we generate 200 samples using CAUKER and calculate a matrix of pairwise Dynamic Time Warping (DTW) distances (Sakoe & Chiba, 1978) on them. We

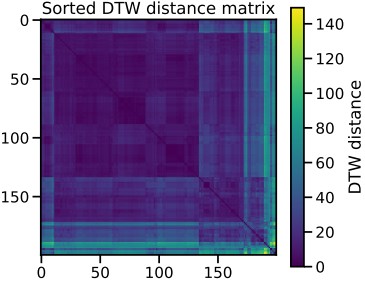

Figure 2: Clustering structure of CAUKER generated dataset with 200 time series.

[1] As the original generator of (Hollmann et al., 2023) is not open–sourced, we followed the algorithmic description in the paper and validated the implementation on the illustrative examples provided therein.

do hierarchical clustering on the obtained precomputed DTW distance matrix and sort the rows and the columns according to the obtained cluster memberships. We plot the obtained matrix in Figure 2. From it, we can observe the emergence of clusters (blocks of time series with similar intra-cluster distances) as well as the introduction of anomalies due to the anomaly mean function in the generating GP. This leads us to believe that CAUKER generates data tailored specifically to classification, which may explain its superiority when pre-training TSFMs on it. More qualitative analysis with SWD (Sliced Wasserstein distance) (Bonneel et al., 2015) and CKNNA (Huh et al., 2024) (classwise $k$-nearest-neighbour alignment) can be find in the Appendix D.

## 4.2 Q2: SCALING LAWS FOR ZERO-SHOT CLASSIFICATION WITH TSFMS

Scaling laws are fundamental to improving foundation models, underpinning their ability to generalize and demonstrate emergent capabilities with increased data and model scale. While scaling laws are widely studied in language and vision, their systematic exploration in the context of zero-shot time series classification remains is currently absent. To the best of our knowledge, our work is the first to thoroughly investigate scaling laws specifically in the setup of zero-shot time series classification which is of independent interest.

### 4.2.1 DATA SCALING LAWS

**Experimental setup**    To investigate data scaling laws, we systematically vary the pre-training dataset sizes from two distinct sources: (1) randomly selected subsets of the real-world UEA benchmark (Bagnall et al., 2018) at increments of $0.1\%$, $1\%$ ... $100\%$, and (2) synthetic data generated by our proposed CAUKER method, at varying scales from 10K up to 10M samples. We recall that both Mantis and MOMENT take as input univariate time series. This means that each channel of multivariate UEA datasets becomes a training sample, with a total of 12M channels (train set and test set combined) from 30 different datasets. Additional details are provided in Appendix E.

**Results**    As illustrated in Figures 3, our experiments indicate that the classification accuracy on the UCR datasets does not monotonically increase with the size of training data when trained on subsets of the UEA dataset (left for Mantis, middle left for MOMENT). We hypothesize that this behavior may be a result of a domain mismatch between UEA and UCR, further exacerbated by the lack of diversity within the real-world time series of UEA. More broadly, we view the irregular UEA scaling as a consequence of how current real-world classification corpora are constructed: UEA aggregates many small, heterogeneous datasets with highly unbalanced sample counts.

In contrast, CAUKER-generated datasets exhibit clear and consistent scaling laws. The accuracy steadily improves with increasing data size, demonstrating the CAUKER-generated data's effectiveness in capturing diverse patterns essential for generalizing to the UCR target set. Additionally, these results also suggest an interesting contrast between model capacities: the lightweight Mantis model achieves competitive performance even with smaller training sets, likely due to the strong time series classification priors incorporated in its architecture that we have mentioned above. In contrast, the larger and more generic MOMENT model exhibits more significant accuracy gains as the training data increases, highlighting its greater capacity to leverage large-scale data for improved representation learning. This distinction underscores the importance of jointly considering model capacity and data availability when designing scalable TSFMs.

### 4.2.2 MODEL SCALING LAWS

**Experimental setup**    We further assessed model scaling laws by varying the size of the MOMENT model (Small, Base, Large versions of sizes 77M, 248M, and 783M, respectively), and Mantis model (with number of parameters 0.75M, 2.59M, 8.10M) using both UEA and CAUKER-generated datasets. More details on the experiments can be found in Appendix F.

**Results**    Results, as shown in Figure 3 (middle right for Mantis, right for MOMENT), indicate that models trained on real-world UEA data do not exhibit consistent performance gains with increasing model size, reinforcing the notion of limited data diversity or domain mismatch. Conversely, models trained on CAUKER-generated datasets consis-

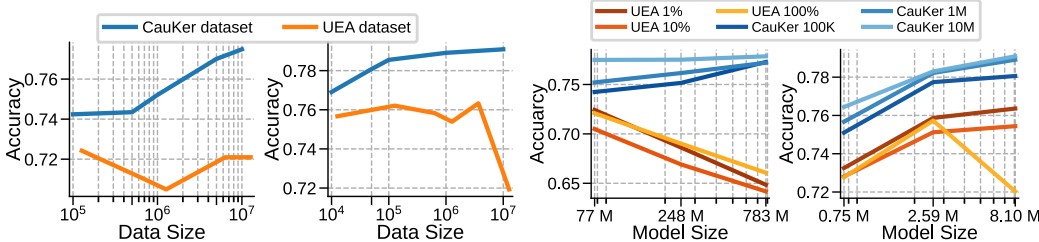

Figure 3: Scaling law of MOMENT and Mantis depending on the dataset size (**left**, **middle left**, respectively) model trained on different subsets of UEA and CauK datasets. Scaling law for the same models depending on the model size (**middle right**, **right**, respectively)

tently demonstrate increased accuracy as model size grows, clearly validating the presence of model scaling laws enabled by the synthetic CAUKER-generated pre-training data.

We further notice that, apart from the single outlier of MOMENT trained on the 10M samples CAUKER corpus, every model pre-trained on CAUKER exhibits a strictly increasing UCR accuracy as its capacity grows. The small increase for MOMENT at 10M indicates that this particular encoder has reached (or is close to) saturation; a similar saturation point can be observed for Mantis once the parameter count exceeds approximately 28M (see Appendix F for a more large-scale experiment). Conversely, the unstable – or even degrading – trend on models pre-trained with larger UEA subsets is most plausibly explained by its lack of diversity. In Figure 4 (and Appendix K), we show PCA projection of CAUKER-generated data, UEA and UCR collections in the embedding space of the original Mantis model. CAUKER-generated data embeddings cover a large region in the embedding space of Mantis fully encompassing both UEA and UCR.

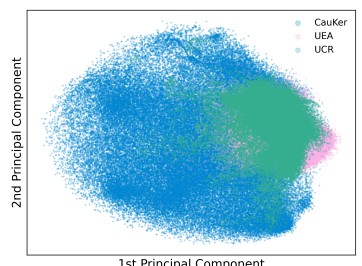

Figure 4: Mantis embeddings of 100K time series drawn from UCR, UEA and generated by CAUKER.

**Qualitative analysis** A recent work by Bouniot et al. (2025) showed that the expressive power of pre-trained vision models can be characterized by measuring their non-linearity. The latter depends not only on the size of the model and its architecture, but also on the pre-training dataset. To verify how TSFMs' expressive power changes depending on the pre-training dataset, we calculate the non-linearity scores of the activation functions inside Mantis as done in the original paper for vision transformers. We then plot the obtained values for the Mantis models pre-trained on CAUKER synthetic datasets of varying sizes and compare them to UEA in Figure 5 (top row). We note that Mantis pre-trained on bigger CAUKER synthetic datasets has a clear trend, while it barely changes when increasing the size

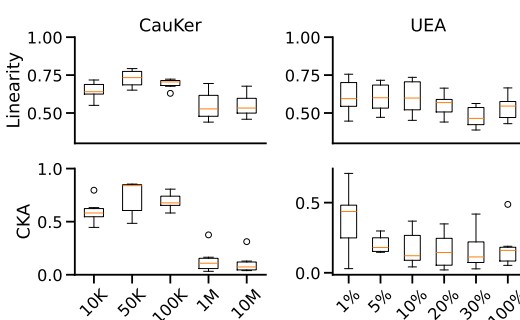

Figure 5: **(Top row)** Non-linearity statistics of the Mantis models pre-trained on CAUKER synthetic datasets of varying size (left) compared to UEA (right); **(Bottom row)** CKA similarities calculated across the hidden layers of the pre-trained models.

of the UEA pre-training sample. Additionally, we validate this finding using the CKA score used to compare the similarity of internal representations of neural networks (Kornblith et al., 2019). Lower values of CKA indicate that the hidden layers change the inputs in a more drastic, non-linear way. We see that pre-training on CAUKER exhibits a structural change in the model's inner workings when the dataset size becomes larger than 100k. In case of real-world UEA data, the CKA scores inside Mantis hidden layers barely change even when the pre-training sample size changes from 600K (5%) to 12M (100%). This, once again, hints at the fact that CAUKER pre-training dataset is much more diverse which aligns well with PCA projection experiment described above.

### 4.3 TRAINING TIME SCALING LAWS

We now study the training time scaling law that aims at identifying the gains in terms of test accuracy that more compute given by longer optimization of the model can bring.

**Experimental setup** We track the evolution of zero-shot accuracy with training epochs for Mantis and MOMENT pre-trained on two corpora, namely a 10% subset of the real-world UEA benchmark and a synthetic set of 1M series generated by CAUKER.

**Results** As illustrated in Figure 6, accuracy rises steadily when the models are trained on CAUKER; additional epochs translate into consistent gains for both architectures. When pre-trained on UEA, however, accuracy curves remain flat or fluctuate, especially for MOMENT, indicating that prolonged optimisation yields little benefit on this dataset. These findings echo the data- and model-scaling observations reported earlier: causally structured, diverse CAUKER data sustains learning over long horizons.

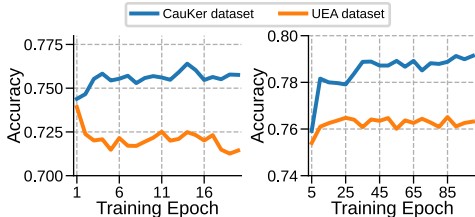

Figure 6: Test accuracy across epochs for MOMENT (**left**) and Mantis (**right**).

### 4.4 Q3: SAMPLE-EFFICIENT PRE-TRAINING OF TSFMS USING CAUKER SYNTHETIC DATA

**Experimental setup** We want to study the performance and the sample efficiency of pre-training Mantis and MOMENT foundation models on different datasets. Our main goal is to show that the performance of both models pre-trained on a total of 1.89M (Mantis) and 13M (MOMENT) unique time series can be almost matched by a pre-training on a smaller synthetic dataset generated using CAUKER. For the latter, we generate as few as 100k samples for Mantis and 10M for MOMENT to account for the model size difference (8M vs. 77M). As before, we include in our study a baseline given by pre-training Mantis and MOMENT on 100k samples of the real-world UEA time series classification collection. Additionally, we also experiment with a subset of 100k time series randomly drawn from standard forecasting datasets (ETTh1, ETTh2, ETTm1, ETTm2, Electricity, ExchangeRate, Illness, Traffic, Weather) (Zhou et al., 2021; Li et al., 2020; Lai et al., 2018; Matsubara et al., 2014; Li et al., 2018; Rasp et al., 2020). Although no prior work trained a classification model on such data, we include it to verify whether the forecasting benchmarks can be a good alternative for classification TSFM pre-training. On average over UCR datasets, classical models trained directly on the raw series achieve 68.12% (Logistic Regression), 69.17% (XGBoost) and 73.25% (Random Forest) accuracy. For completeness, we also report downstream fine-tuning results in Appendix M.

| Model | pre-train. set | Size | UCR Included? | UCR acc. (%) |
|---|---|---|---|---|
| Mantis | CAUKER | 100K | No | 78.55 |
| | Mantis dataset | 1.89M | Yes | 78.66 |
| | UEA | 100K | No | 76.73 |
| | Forecasting | 100K | No | 75.81 |
| MOMENT | CAUKER | 10M | No | 77.49 |
| | Time Series Pile | 13M | Yes | 78.85 |
| | CAUKER | 100K | No | 74.24 |
| | UEA | 100K | No | 73.55 |
| | Forecasting | 100K | No | 73.93 |

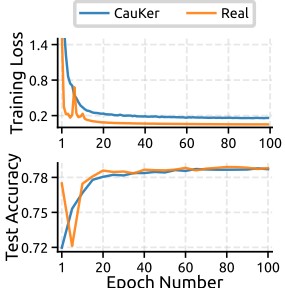

Figure 7: Performance comparison of Mantis and MOMENT models on different pre-training datasets. CAUKER-generated pre-training data allows to nearly match the performance of the original TSFMs, while being more sample-efficient. Rows with 'UCR included? = Yes' correspond to in-distribution zero-shot evaluation, as the pre-training corpus contains UCR train splits (though not test data). Rows with 'UCR included? = No' correspond to strictly OOD zero-shot models. Training loss and test accuracy corresponding to the first two rows illustrated in the right figure show that synthetic data is harder to train on, but leads to a smoother increase of the test accuracy across epochs.

**Results** From the results presented in Table 7, we note that the performances of Mantis and MOMENT can be almost matched by pre-training them on synthetic datasets that are $\sim 20\times$ and $\sim 1.3\times$ smaller than the original pre-training datasets used by each of the papers. The accuracy drop in the case of Mantis is less than 0.1%, while for MOMENT it barely exceeds 1%. This suggests that the synthetic data generated by CAUKER makes model pre-training more sample-efficient. An additional evaluation on 17 datasets from real-world WOODS benchmark (Gagnon-Audet et al., 2023) comparing original Mantis and Mantis pre-trained on 100K CAUKER time series confirms this finding (see Appendix I). Beyond UCR and WOODS, we further evaluate CAUKER pre-trained models on irregular, multivariate clinical benchmarks from (Li et al., 2023b), and find that they remain competitive with the original model in irregular settings as well, see Appendix N for details. We also note that the training loss and test accuracy of Mantis pre-trained on 100k and 1.89M time series exhibit a very different behavior. For the synthetic dataset, the training loss remains higher, indicating that it is harder to learn, likely due to the high diversity of the generated time series. Yet, the test accuracy in this case steadily improves and surpasses the accuracy of the original model, which quickly learns the real-world pre-training dataset. This is reminiscent of the MOMENT pre-training, which only required 2 epochs (Goswami et al., 2024) (even for the largest 783M) to converge.

In addition to this, the reported UCR classification accuracies of the original Mantis and MOMENT models represent *in-distribution* performance, since their respective training corpora include UCR train samples. In this sense, these scores may serve as a practical upper bound for zero-shot accuracy, beyond which out-of-distribution generalization is unlikely without direct exposure to test distributions. Finally, we note that the comparison with two other pre-training dataset candidates leads to strictly worse results.

**Extension to forecasting.** Although our primary contribution focuses on classification, we also experimented with applying CAUKER to forecasting TSFM. Interestingly, we observed that our pipeline transfers effectively to the forecasting setting as well: Chronos models (tiny, mini, small and base) pre-trained exclusively on 0.5B timepoints CAUKER-generated series achieve zero-shot forecasting accuracy that is statistically indistinguishable from the original models pre-trained on 84B tokens (p-value of 0.84 at the significance level of 0.05 for two-sided Wilcoxon signed rank test). We provide details and results in Appendix J. Importantly, this result is obtained without any task-specific modifications to the CAUKER pipeline.

## 5 CONCLUSION

In this work, we introduced CAUKER, a novel synthetic data generation framework tailored for time series classification. By integrating Gaussian Process kernel composition with Structural Causal Models, CAUKER generates synthetic datasets that are both temporally realistic and causally coherent. We demonstrated that TSFMs pre-trained solely on CAUKER-generated data can match the performance of models trained on larger real-world datasets. Furthermore, our study provides the first in-depth analysis of data and model scaling laws in zero-shot time series classification, establishing that such scaling effects emerge clearly when using synthetic data, but are irregular or absent when training on commonly used real-world datasets.

Our findings underscore a key insight already known in vision and natural language processing: the quality and structure of pre-training data have a profound impact on the generalization performance of TSFMs. While much recent progress in time series community has focused on architectural innovations, our results suggest that equivalent gains can be achieved through principled design of synthetic training data. We hope this work encourages the community to direct greater attention to the design, analysis, and benchmarking of time series training datasets, as a complementary path toward building scalable, general-purpose time series foundation models.

**Limitations** Similar to prior work on scaling laws in time series forecasting (Yao et al., 2025), we considered only two models that follow a different pre-training paradigm. As our study was already quite compute-intense, we believe that this choice is justified. In the same line, we didn't consider large-scale forecasting benchmarks such as Time-300B (Shi et al., 2025) as we have observed that forecasting benchmarks are of limited utility for classification.

ACKNOWLEDGEMENTS

Supported by EU project DataGEMS (101188416), and by $Y\Pi AI\Theta A$ & NextGenerationEU project HARSH ($Y\Pi 3TA - 0560901$) that is carried out within the framework of the National Recovery and Resilience Plan "Greece 2.0" with funding from the European Union – NextGenerationEU. This work was granted access to the HPC resources of IDRIS under the allocation 2025-A0191012641 made by GENCI.

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

## APPENDIX

The rest of this appendix is organized as follows. In Section A, we provide an overview of pre-training datasets commonly used in TSFMs. In Section B, we describe the contrastive and masked learning losses as well as architectural details of representative models Mantis and MOMENT. Section C introduces the CAUKER pipeline in detail, including pseudocode, kernel / mean / activation banks, and hyperparameter sensitivity analysis. Section D presents additional qualitative analyses on global and local alignment between synthetic and real data. Experimental details for data scaling laws are provided in Section E, while model scaling experiments are discussed in Section F. Further training and evaluation details are summarized in Section G. Section H reports a domain-wise UCR breakdown, and Section I extends evaluation to the WOODS benchmark. Section J presents results of pre-training Chronos on CAUKER and zero-shot forecasting. Visualization analyses of embeddings are discussed in Section K. Finally, we clarify the use of Large Language Models for writing assistance in the last section.

## A    OVERVIEW OF PRE-TRAINING DATASETS FOR TIME SERIES FOUNDATION MODELS

Table 3 summarizes the pre-training datasets used by representative Time Series Foundation Models. For each model, we report whether synthetic data was used, the total number of time points and time series samples, whether the datasets are publicly available. The table is organized alphabetically by model name.

| Model | Synthetic | Real | Time Points | Series Count | Open |
|---|---|---|---|---|---|
| Chronos (Ansari et al., 2024) | Yes | Yes | 84B | 890K | Yes |
| ForecastPFN (Dooley et al., 2023) | Yes | No | 60M | 300K | Yes |
| Mantis (Feofanov et al., 2025) | No | Yes | N/A | ~1.89M [1] | Yes |
| MOMENT(Goswami et al., 2024) | No | Yes | 1.23B | 13M | Yes |
| NuTime (Lin et al., 2024) | No | Yes | 60M | 1.89M | Yes |
| TabPFN (Hollmann et al., 2023) | Yes | NO | N/A | 9.216M | No |
| TimePFN (Taga et al., 2025) | Yes | No | $\sim 200M$ | ~3M | Yes |
| UniTS (Gao et al., 2024) | No | Yes | 35M | 6K | Yes |

Table 3: Overview of pre-training datasets for Time Series Foundation Models (TSFMs).

## B    LOSS AND ARCHITECTURE OF MANTIS AND MOMENT

**Contrastive learning loss of Mantis.**    Given an encoder $F : \mathbb{R}^t \to \mathbb{R}^q$, we consider random augmentations $\phi, \psi \sim \mathcal{U}(\mathcal{T})$. The similarity between two augmented samples is measured after projecting their embeddings to a new dimension $q'$ via $g : \mathbb{R}^q \to \mathbb{R}^{q'}$. Specifically, the cosine similarity is defined as:

$$s_{\cos}(\mathbf{a}, \mathbf{b}) = \frac{\mathbf{a}^\top \mathbf{b}}{\|\mathbf{a}\|\|\mathbf{b}\|}, \quad \forall(\mathbf{a}, \mathbf{b}) \in \mathbb{R}^{2q'}.$$

Given a batch $B = \{\mathbf{x}_i\}_{i=1}^b$, we compute pairwise similarities:

$$\mathbf{s}_i(\phi, \psi) = [s_{\cos}(g \circ F \circ \phi(\mathbf{x}_i), g \circ F \circ \psi(\mathbf{x}_j))]_{j=1}^b \in \mathbb{R}^b.$$

The Mantis encoder $F$ and projector $g$ are optimized by minimizing the contrastive loss:

$$\mathcal{L}_{\text{contrastive}} = \sum_{i=1}^b l_{\text{ce}}\left(\frac{\mathbf{s}_i(\phi, \psi)}{T}, i\right),$$

where $l_{\text{ce}}$ is the cross-entropy loss and $T$ is a temperature parameter set to 0.1.

**Masked learning loss of MOMENT.** Given a univariate time series $\mathcal{T} \in \mathbb{R}^{1 \times T}$, it is segmented into $N$ disjoint patches of length $P$. Each patch is mapped into a $D$-dimensional embedding, replaced with a learnable mask embedding $[\text{MASK}] \in \mathbb{R}^{1 \times D}$ for masked patches. The resulting embeddings are fed into a transformer encoder, producing transformed embeddings that are then decoded by a lightweight reconstruction head $h_{\text{rec}}$. The masked loss for reconstruction is defined as the mean squared error (MSE):

$$\mathcal{L}_{\text{masked}} = \frac{1}{|\Omega|} \sum_{n \in \Omega} \|\mathcal{T}_n - h_{\text{rec}}(F([\text{MASK}]))_n\|^2,$$

where $\Omega$ denotes the set of indices corresponding to masked patches.

**Model architectures.** For the masked learning approach, MOMENT leverages a Transformer-based architecture derived from the T5 family (Chung et al., 2022)model. Specifically, MOMENT employs a 8, 12, 24-layer Transformer encoder with hidden dimensions $D = 512, 768, 1024$, and 8, 12, 16 attention heads for "Small", "Base", "Large" model. The model processes input time series by segmenting them into $N = 64$ patches of length $P = 8$, applying positional embeddings, and then reconstructing masked patches.

Conversely, Mantis utilizes a Vision Transformer (ViT)(Dosovitskiy et al., 2021) architecture. Initially, the input time series is divided into tokens, to which a learnable class token is appended. Positional embeddings are added to encode temporal information explicitly. The ViT unit consists of 6 transformer layers, each comprising multi-head attention with 8 heads. The final output is derived from the class token's embedding after aggregation by the transformer layers. It is worth noting that Mantis employs a customized tokenizer. For detailed information, please refer to the original Paper[2].

## C DETAILS OF CAUKER

### C.1 PSEUDOCODE OF THE CAUKER

Algorithm 1 describes the full synthetic data generation process of CAUKER. The pipeline combines the temporal structure modeled by Gaussian processes with the flexible dependency modeling of structural causal models. Specifically, the algorithm first samples a number of root signals from GP priors constructed via randomly composed kernels and mean functions. It then propagates these signals through a randomly generated DAG, where each edge applies a nonlinear transformation drawn from an activation function bank. Finally, a fixed number of node outputs are selected as observed time series variables, each interpolated to a target length. This modular and stochastic design ensures rich diversity and causal consistency in the generated synthetic data.

### C.2 DETAILS OF BANKS

Figure 8 provides illustrative examples of the six representative kernels selected from our base kernel bank. The top row of the figure displays the covariance matrices induced by each kernel over 1024 evenly spaced time points, while the bottom row shows corresponding sample paths drawn from the Gaussian Process (GP) prior using these kernels.

Specifically, the illustrated kernels include:

---

[1]The updated number of training samples ($\sim$1.38M) is confirmed in the official repository: `https://github.com/vfeofanov/mantis/issues/2`. The arXiv version initially reported $\sim$7M.

[2]`https://github.com/vfeofanov/Mantis`

---

**Algorithm 1:** CAUKER: Synthetic Time–Series Generator for Classification

---

**Input:** $N$ ;      `// total number of samples to output`
  1   $L$ ;      `// target length of each time series`
  2   $d$ ;      `// number of observed variables per sample`
  3   Banks $\mathcal{K}$ (kernels), $\mathcal{M}$ (mean fns), $\mathcal{A}$ (activations)
  4   Hyper–parameters: $K_{\max}, V_{\max}, P_{\max}$ ;    `// max kernels, nodes, parents`
  5   RNG ;      `// random generator with fixed seed`
**Output:** $\mathcal{D}_{\text{syn}} = \{x_1, \ldots, x_N\}, \; x_i \in \mathbb{R}^{d \times L}$
6 **Function** SAMPLECOMPOSITEKERNEL($\mathcal{K}$)**:**
7      $K \leftarrow \text{RNG.UniformInt}(1, K_{\max})$;
8      $\kappa \leftarrow \text{RNG.Choice}(\mathcal{K})$;
9      **for** $i \leftarrow 2$ **to** $K$ **do**
10         $\kappa_i \leftarrow \text{RNG.Choice}(\mathcal{K})$;
11         op $\leftarrow \text{RNG.Choice}(\{+, \times\})$;
12         $\kappa \leftarrow \kappa \text{ op } \kappa_i$;
13      **return** $\kappa$;
14 **Function** SAMPLEMEAN($\mathcal{M}$, $x$)**:**
15      $m_1, m_2 \leftarrow \text{RNG.Choice}(\mathcal{M}, \text{size} = 2, \text{replace} = True)$;
16      op $\leftarrow \text{RNG.Choice}(\{+, \times\})$;
17      **return** $\text{op}(m_1(x), m_2(x))$;

18 $\mathcal{D}_{\text{syn}} \leftarrow \emptyset$;
19 **while** $|\mathcal{D}_{\text{syn}}| < N$ **do**
20      $V \leftarrow \text{RNG.UniformInt}(d, V_{\max})$;
21      $G = (\mathcal{V}, \mathcal{E}) \leftarrow \text{RANDOMDAG}(V, P_{\max})$ ;
22      $roots \leftarrow \{v \in \mathcal{V} \mid \deg^-(v) = 0\}$;
23      $\Sigma \leftarrow \text{RNG.Choice}(\mathcal{A}, \text{size} = |\mathcal{E}|, \text{replace} = True)$;
24      map each $e \in \mathcal{E}$ uniquely to an activation $\sigma_e \in \Sigma$;
25      **foreach** $r \in roots$ **do**
26         $\kappa_r \leftarrow \text{SAMPLECOMPOSITEKERNEL}(\mathcal{K})$ ;
27         $\mu_r(\cdot) \leftarrow \text{SAMPLEMEAN}(\mathcal{M}, \cdot)$ ;
28         $t_r \sim \mathcal{GP}(\mu_r, \kappa_r)$ on grid $[0, 1]$ of length $L$;
29      **foreach** $v \in \text{TopoSort}(G)$ **do**
30         **if** $v \in roots$ **then**
31             **continue**
32         $P_v \leftarrow \{u \mid (u, v) \in \mathcal{E}\}$;
33         $\mathbf{z} \leftarrow \text{Concat}(\{t_u\}_{u \in P_v})$;
34         $W \sim \mathcal{N}(0, 1)^{1 \times |P_v|}, \quad b \sim \mathcal{N}(0, 1)$;
35         $t_v \leftarrow \sigma_{(u,v)}(W\mathbf{z} + b)$;
36      $\mathcal{V}' \leftarrow \text{RNG.Choice}(V, size{=}d, replace{=}False)$   x$\leftarrow \text{Stack}(\{t_v\}_{v \in \mathcal{V}'})$;
37      $\mathcal{D}_{\text{syn}} \leftarrow \mathcal{D}_{\text{syn}} \cup \{x\}$;
38 **return** $\mathcal{D}_{\text{syn}}$

---

- **ExpSineSquared** — captures periodic patterns with a fixed wavelength; produces strongly oscillatory samples with global smoothness.

- **DotProduct** — induces linear trend behavior; sample paths grow or decay steadily over time.

- **RBF (Radial Basis Function)** — generates smooth, localized fluctuations around zero with short-range correlations.

- **RationalQuadratic** — a scale mixture of RBF kernels, allowing for multiscale smooth variations in the signal.

- **WhiteKernel** — models uncorrelated noise; sample paths resemble pure Gaussian noise with no temporal structure.

- **ConstantKernel** — generates flat constant signals; serves as a component for additive models with nonzero mean.

These six kernels represent only a small subset of our full kernel bank. In practice, we construct a much larger kernel bank comprising 36 distinct kernels. This is achieved by varying the hyperparameters of each kernel (e.g., length-scale, periodicity, noise level, amplitude) across a range of scales to capture diverse temporal dynamics. For instance, we use multiple versions of the ExpSineSquared kernel with different periodicities to simulate both high- and low-frequency periodic patterns. Similarly, we vary the length-scale of RBF and RationalQuadratic kernels to control smoothness and correlation range. An important point is that CAUKER randomly samples a small number of kernels from the bank for each composite GP, so enlarging or slightly modifying the bank does not increase computational cost.

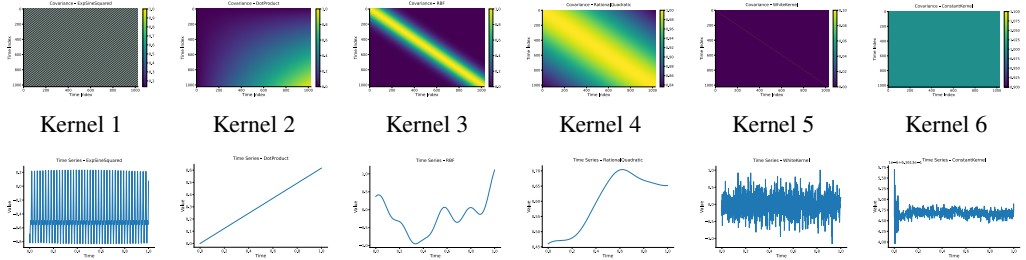

Figure 8: Visualizations of covariance matrices (top) and corresponding sampled time series (bottom) from each base kernel in the kernel bank.

The images presented in Figure 8 serve as illustrative examples only. During synthetic data generation, kernels are sampled from the full kernel bank, which offers significantly richer diversity than what is shown here. These base kernels are subsequently composed using random additive and multiplicative operations to define flexible Gaussian process priors for root node generation in the CAUKER pipeline.

Figure 9 presents the four representative mean functions used in our synthetic data generation pipeline. Each subplot illustrates a randomly sampled instance from the corresponding function class. These functions can be combined multiplicatively or additively during Gaussian process sampling to enrich the diversity of generated signals.

- Zero Mean: A baseline function returning a constant zero across the time axis, corresponding to the standard GP assumption with zero-centered priors.
- Linear Mean: A simple affine transformation $a \cdot t + b$, enabling trends such as monotonic increases or decreases over time.
- Exponential Mean: A parametric form $a \cdot \exp(bt)$ that introduces strong, nonlinear growth or decay patterns into the signal.
- Sparse Anomalies: A piecewise-constant mean vector with a few randomly placed spikes, simulating rare disruptive events (e.g., faults, attacks, regime shifts).

These mean functions serve as building blocks for composing realistic non-stationary temporal structures in synthetic time series. In the generation process, two functions are randomly selected and combined (either by summation or elementwise multiplication), forming the final mean vector used in GP sampling. The images shown in Figure 9 are illustrative samples; in practice, stochastic variation over parameters (slopes, amplitudes, etc.) ensures that each generated series presents unique mean behavior.

**Activation function bank.** In addition to kernel and mean banks, CAUKER employs a diverse *activation function bank* $\mathcal{A}$ to propagate nonlinear transformations through the structural causal graph. Each edge in the DAG is randomly assigned an activation from this bank, which governs how parent node values influence their children. The activation bank comprises both classical and domain-specific transformations:

- **Linear:** Identity or affine mappings $ax + b$, preserving proportional signal propagation.

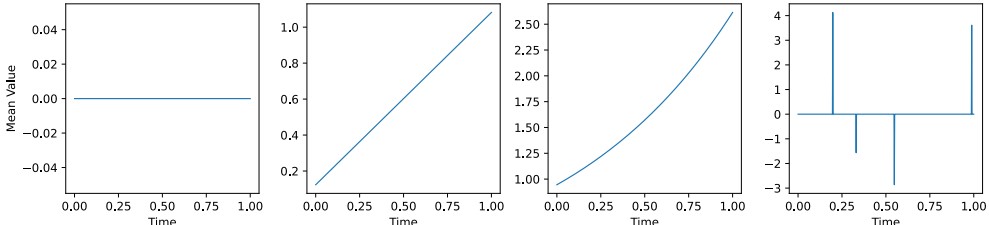

Figure 9: Examples of four mean function types used in the synthetic data pipeline. Each function introduces distinct temporal structure, contributing to the diversity and realism of generated sequences.

- **ReLU:** Rectified linear units $\max(0, x)$, introducing sparsity and piecewise linearity.
- **Sigmoid:** Smooth squashing function $\sigma(x) = 1/(1 + e^{-x})$, modeling saturation effects.
- **Sinusoidal:** Periodic modulations $\sin(x)$, inducing wave-like behaviors.
- **Modulo:** Modular transformations $x \bmod c$, yielding abrupt nonlinearities or periodic clipping.
- **Leaky ReLU:** Slope-preserving variant of ReLU, ensuring non-zero gradients for negative inputs.

These nonlinearities enhance the diversity of functional relationships within the generated synthetic time series and allow the resulting signals to exhibit complex, structured dependencies. As illustrated in the SCM pipeline, these functions are applied edge-wise to linear combinations of parent signals before assigning values to child nodes.

## C.3 HYPERPARAMETER SENSITIVITY OF CAUKER

We report an ablation on two families of CAUKER hyperparameters that control the generative complexity of synthetic series: (i) a co-sweep where we simultaneously increase the number of randomly sampled kernels used in the composite GP prior ("Kernel $k$") and the maximum number of parents per node in the DAG ("Parent $p$"), and (ii) the graph size (number of nodes) of the DAG. [3]

**Observations.** Along the Kernel/Parents co-sweep, the data (shown in 4)statistics exhibit consistent trends: Entropy, Stability, and Lumpiness increase steadily, while the Hurst decreases, indicating more heterogeneous and less persistent series as complexity grows. Despite these changes in data characteristics, the downstream UCR Accuracy remains essentially stable (fluctuations within $\approx 0.4\%$). When varying the graph size, the method is likewise insensitive (shown in 5): the UCR accuracy varies within a narrow band, while the data statistics change only mildly. Overall, CAUKER is robust to these generative hyperparameters.

**Failure modes** To probe failure modes, we force CAUKER to use only a single kernel family (plus SCM). DotProduct-only GPs, which produce almost linear trends, lead to substantially worse performance (76.79% UCR accuracy). In contrast, RBF-only sampling remains competitive (78.07%), underscoring the importance of sufficiently rich nonlinear structure and kernel diversity.

## D ADDITIONAL QUALITATIVE ANALYSIS

To complement the qualitative discussion in §4.1, we quantify how well synthetic corpora produced by different generators resemble the target UCR distribution, both globally and locally. Concretely, we report:

---

[3]Entropy, Hurst, Stability, and Lumpiness are computed with the same definitions in (Aksu et al., 2024); higher Stability and Lumpiness indicate stronger regime structure/heterogeneity, while larger Hurst indicates more persistent (long-memory) behavior.

Table 4: **Kernel/Parents co-sweep.** Increasing both the number of sampled kernels in the GP composition and the maximum number of parents per node produces steadily higher Entropy/Stability/Lumpiness and a decreasing Hurst, while UCR accuracy stays stable.

| Kernel / Parents | Entropy | Hurst | Stability | Lumpiness | UCR Acc. |
|---|---|---|---|---|---|
| Kernel3 / Parent2 | 0.4629 | 0.7719 | 0.9821 | 145.18 | 0.7848 |
| Kernel4 / Parent3 | 0.5034 | 0.7713 | 1.7949 | 1081.17 | 0.7854 |
| Kernel5 / Parent4 | 0.5352 | 0.7686 | 2.8164 | 6348.75 | 0.7850 |
| Kernel6 / Parent5 | 0.5522 | 0.7655 | 4.5419 | 1854961.61 | 0.7825 |
| Kernel7 / Parent6 | 0.6225 | 0.7519 | 11.7237 | 10148441.78 | 0.7810 |

Table 5: **Graph size sweep.** CAUKER is insensitive to DAG size.

| Graph Size | Entropy | Hurst | Stability | Lumpiness | UCR Acc. |
|---|---|---|---|---|---|
| 10 | 0.5273 | 0.7707 | 2.0424 | 2314.14 | 0.7848 |
| 20 | 0.5716 | 0.7664 | 3.7253 | 1142.36 | 0.7811 |
| 30 | 0.5714 | 0.7665 | 2.1988 | 1252.20 | 0.7812 |
| 40 | 0.5818 | 0.7630 | 2.1085 | 836.93 | 0.7815 |
| 50 | 0.5889 | 0.7654 | 2.3601 | 50469.86 | 0.7785 |

- **Global proximity** via the Sliced Wasserstein distance (SWD). Given empirical distributions $P$ and $Q$ over $\mathbb{R}^L$, the $2^{\text{nd}}$-order SWD is

$$\text{SWD}_2(P, Q) = \mathbb{E}_{\theta \sim \mathcal{U}(\mathbb{S}^{L-1})} \, W_2\big(\langle \theta, X \rangle, \langle \theta, Y \rangle\big),$$

where $W_2$ is the one-dimensional 2-Wasserstein distance between the projected marginals, and $\theta$ is sampled uniformly on the unit sphere.

- **Local alignment** via CKNNA (classwise $k$-nearest-neighbour alignment) following Huh et al. (2024). Let $f$ be a frozen encoder and let $\mathcal{S}$ (source, synthetic pre-training) and $\mathcal{T}$ (target, UCR) denote labelled sets. For each $x \in \mathcal{T}$ with label $y_x$, we form the $k$-NN set $\mathcal{N}_k(x; f(\mathcal{S}))$ under cosine distance in feature space and define

$$\text{CKNNA}_k = \frac{1}{|\mathcal{T}|} \sum_{x \in \mathcal{T}} \frac{1}{k} \sum_{z \in \mathcal{N}_k(x; f(\mathcal{S}))} \mathbb{I}\big[y_z = y_x\big],$$

then average over datasets.

For each UCR dataset, we compare it to five independent synthetic draws produced by (i) KER-NELSYNTH and (ii) CAUKER, using the same generator hyperparameter priors as in §4.1. Global proximity is computed by averaging $\text{SWD}_2$ over 512 random one-dimensional projections per dataset; local alignment uses features from a frozen encoder and classwise $k$-NN agreement as above. We report mean $\pm$ standard deviation across the five synthetic draws and then average across UCR datasets.

Table 6: **Global and local alignment between UCR and synthetic corpora.** Lower is better for SWD; higher is better for CKNNA. Means $\pm$ s.d. across five independent synthetic draws, then averaged over UCR datasets.

| | KernelSynth | CAUKER |
|---|---|---|
| Global SWD$_2$ | 7.11 ($\pm$1.13) | **3.1486 ($\pm$0.21)** |
| CKNNA (avg. over datasets) | 0.014 ($\pm$0.03) | **0.015 ($\pm$0.03)** |

**Findings.** CAUKER achieves substantially smaller global discrepancy to UCR than KERNELSYNTH (Table 6, $SWD_2$), indicating a closer match at the dataset-level distribution. At the same time, its (slightly) higher CKNNA suggests at least comparable—and typically better—classwise local neighbourhood structure transfer from pre-training to UCR. Together, these results support the view that the SCM backbone in CAUKER is crucial for shaping both global statistics and local class geometry in ways that benefit zero-shot classification.

# E  EXPERIMENTAL DETAILS OF SECTION 4.2.1

In our scaling law experiments, we systematically evaluated the performance of two distinct models, Mantis and MOMENT, across varying dataset sizes from both real-world and synthetic sources. We adopted the official 8M parameters configuration of Mantis as released in its open-source repository, which includes a 6-layer ViT encoder with 8 attention heads and a hidden dimension of 256. The classification head used was a Random Forest classifier trained on frozen embeddings.

For MOMENT, we used the officially supported "google/flan-t5-small" variant containing 77M parameters as the encoder backbone. This model structure is one of the pre-trained configurations endorsed in the original MOMENT framework. During training, we froze the encoder and trained only the classification head, which was implemented as a Support Vector Machine (SVM). This setup mirrors the zero-shot classification evaluation protocol used in prior TSFM literature.

For both models, we varied the training data sizes as follows: for the real-world UEA dataset, subsets ranging from 0.1% to 100% (12.7K to 12.67M samples) were randomly sampled. For synthetic data, we generated samples using our CAUKER method at 10K, 50K, 100K, 500K, 1M, 5M, and 10M scales. All series were univariate with length 512. The full list of data sizes and corresponding classification accuracy values on the UCR benchmark are reported in Table 7.

| Model | Train Set | Data Size | UCR Accuracy (%) |
|---|---|---|---|
| MOMENT (77M) | UEA | 127K | 72.42 |
| | UEA | 1.27M | 70.49 |
| | UEA | 633K | 71.09 |
| | UEA | 6.33M | 72.09 |
| | UEA | 12.67M | 72.10 |
| | CAUKER | 100K | 74.24 |
| | CAUKER | 500K | 74.35 |
| | CAUKER | 1M | 75.21 |
| | CAUKER | 5M | 77.01 |
| | CAUKER | 10M | 77.49 |
| Mantis (8M) | UEA | 12.7K | 75.67 |
| | UEA | 127K | 76.21 |
| | UEA | 633K | 75.83 |
| | UEA | 1.27M | 75.39 |
| | UEA | 3.68M | 76.33 |
| | UEA | 12.67M | 71.93 |
| | CAUKER | 10K | 76.91 |
| | CAUKER | 50K | 78.08 |
| | CAUKER | 100K | 78.55 |
| | CAUKER | 1M | 78.91 |
| | CAUKER | 10M | 79.09 |

Table 7: Exact accuracy values used in the scaling law plots (Figure 3).

## F    EXPERIMENTAL DETAILS OF SECTION 4.2.2

To investigate model scaling laws, we evaluated a range of model capacities for both MOMENT and Mantis using synthetic datasets generated by CAUKER. For MOMENT, we adopted the official series of models given by:

- **flan-t5-small** (77M parameters),
- **flan-t5-base** (248M parameters),
- **flan-t5-large** (783M parameters).

For the Mantis encoder, we varied the transformer depth and width while keeping the sequence length fixed at 512 and using the same patching configuration. The model variants are as follows:

- **0.75M**: `hidden_dim=256`, `transf_depth=1`, `transf_num_heads=2`, `transf_mlp_dim=512`, `transf_dim_head=128`.
- **2.59M**: same as above, with `transf_depth=3`, `transf_num_heads=4`.
- **8.10M**: same as above, with `transf_depth=6`, `transf_num_heads=8`.
- **28.56M**: same as above, with `transf_depth=12`, `transf_num_heads=16`.
- **114.14M**: `hidden_dim=512`, `transf_depth=12`, `transf_num_heads=16`, `transf_mlp_dim=1024`, `transf_dim_head=256`.

All Mantis variants used the following fixed parameters: `seq_len=512`, `num_patches=32`, `scalar_scales=None`, `hidden_dim_scalar_enc=32`, and `epsilon_scalar_enc=1.1`. The model output embeddings were classified using a Random Forest classifier trained on frozen features.

This design allows us to jointly assess the impact of model depth, width, and hidden dimensionality on zero-shot classification performance under a consistent synthetic data regime.

Table 8 reports the exact accuracy values corresponding to the model scaling plots shown in Figure 10. For both MOMENT and Mantis, we list results under varying model sizes and dataset configurations.

| Model Size | UEA 1% | UEA 10% | UEA 100% | CAUKER 100K | CAUKER 1M | CAUKER 10M |
|---|---|---|---|---|---|---|
| 77M (MOMENT) | 72.42 | 70.49 | 72.10 | 74.24 | 75.21 | 77.49 |
| 248M (MOMENT) | 68.62 | 66.91 | 69.01 | 75.16 | 76.16 | 77.51 |
| 783M (MOMENT) | 64.85 | 64.18 | 66.07 | 77.28 | 77.20 | 77.85 |
| 0.75M (Mantis) | 73.25 | 72.81 | 72.77 | 75.10 | 75.67 | 76.44 |
| 2.59M (Mantis) | 75.87 | 75.12 | 75.73 | 77.74 | 78.22 | 78.30 |
| 8.10M (Mantis) | 76.36 | 75.44 | 72.03 | 78.06 | 78.91 | 79.09 |
| 28.56M (Mantis) | 76.66 | 77.15 | 77.05 | 78.70 | 78.83 | 78.19 |
| 114.14M (Mantis) | 76.60 | 77.29 | 76.97 | 78.42 | 78.86 | 78.81 |

Table 8: Exact zero-shot accuracy (%) on the UCR benchmark under different model sizes and pre-training dataset configurations.

## G    EXPERIMENTAL DETAILS OF SECTION 4.4

For all compared models, we adopted the best training loss epoch as the checkpoint for final evaluation. Specifically, the official setting for Mantis involves training for 100 epochs, while MOMENT is typically trained for 2 epochs. However, for our experiments, we trained Mantis for 100 epochs and MOMENT for 10 epochs to allow sufficient convergence, consistent with our goal of achieving the best performance on the CAUKER and UEA datasets. For the MOMENT model, we utilized the base model "google/flan-t5-small" with 77M parameters, trained on both the CAUKER and UEA datasets. The official MOMENT checkpoint used in our experiments (Time Series Pile), "google-t5/t5-small," has 60M parameters.

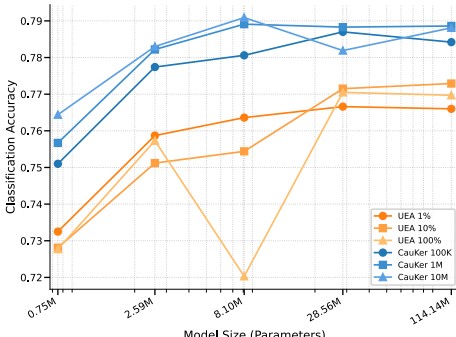

Figure 10: Accuracy on UCR dataset with varying model sizes for the Mantis model trained on UEA subsets and synthetic CAUKER data.

## H    DOMAIN-WISE ANALYSIS ON UCR

To assess whether the gains observed with CAUKER pre-training are uniform across application areas, we group the UCR datasets (Dau et al., 2019) by the standard Type taxonomy (e.g., ECG, Motion, Spectro, etc.) and report domain-wise averages of zero-shot accuracies. We compare (i) a Mantis encoder pre-trained on 100K CAUKER synthetic series (CauKer100K) against (ii) the official Mantis checkpoint released by the authors.[4] For each domain we also report the absolute difference $\Delta = \text{CauKer100K} - \text{Official}$.

Table 9: Domain-wise UCR accuracy (mean across datasets within each Type). $\Delta = \text{CauKer100K} - \text{Official}$.

| Type | CauKer100K | Official | $\Delta$ |
|------|------------|----------|----------|
| Device | 0.7209 | 0.7288 | $-0.0079$ |
| ECG | 0.8539 | 0.8271 | $+0.0268$ |
| EOG | 0.5000 | 0.5304 | $-0.0304$ |
| EPG | 0.9980 | 1.0000 | $-0.0020$ |
| HRM | 0.8602 | 0.8387 | $+0.0215$ |
| Hemodynamics | 0.7131 | 0.7179 | $-0.0048$ |
| Image | 0.7798 | 0.7910 | $-0.0112$ |
| Motion | 0.7883 | 0.7873 | $+0.0010$ |
| Power | 0.9667 | 0.9056 | $+0.0611$ |
| Sensor | 0.7823 | 0.7913 | $-0.0091$ |
| Simulated | 0.9434 | 0.9355 | $+0.0078$ |
| Spectro | 0.7226 | 0.7775 | $-0.0550$ |
| Spectrum | 0.8126 | 0.7961 | $+0.0165$ |
| Traffic | 0.9120 | 0.8869 | $+0.0251$ |
| Trajectory | 0.5385 | 0.5282 | $+0.0103$ |

**Results and discussion**    Eight out of fifteen domains exhibit higher accuracy with CAUKER pre-training (Table 9). The only domain showing a substantial degradation (greater than $5\%$) is Spectro $(-5.50\%)$. A plausible explanation is the prevalence of very small spectroscopy datasets (often $< 50$ samples), which amplifies the in-distribution advantage of the Official Mantis model that was exposed to UCR training splits during pre-training.

---

[4]The official Mantis checkpoint was pre-trained on a corpus that includes the UCR training splits; thus, its reported scores constitute in-distribution performance with a potential advantage on small domains.

Table 10: WOODS summary (domain averages over constituent datasets) and overall statistics. ERM: supervised baseline from Gagnon-Audet et al. (2023). Mantis-2M: original real-data pre-trained encoder ($\sim$1.89M series). CauKer100K: the same architecture pre-trained on 100K CAUKER samples.

| Domain / Statistic | ERM | CauKer100K | Mantis-2M |
|---|---|---|---|
| CAP (EEG) | 0.750 | **0.782** | 0.760 |
| HAR | 0.934 | **0.946** | 0.940 |
| MI (EEG) | **0.733** | 0.563 | 0.543 |
| SEDFx (EEG) | 0.7225 | **0.7700** | 0.7375 |
| Win counts (out of 17; ties counted) | 7 | **11** | 4 |
| Average over all 17 datasets | 0.800 | **0.820** | 0.810 |

Conversely, the largest improvement is observed on the Power domain ($+6.11\%$), consistent with the presence of strong periodic and quasi-seasonal motifs in power consumption profiles that are well captured by the Gaussian-process kernel composition within the CAUKER pipeline. Across the remaining thirteen domains, the performance gap remains modest (within $\pm3\%$), indicating that CAUKER-based pre-training transfers robustly across diverse application areas despite using only 100K synthetic series.

These findings support our main claim: causally structured, kernel-composed synthetic pre-training yields competitive (and sometimes superior) representations for zero-shot classification across heterogeneous time-series domains, while the rare large deficit (here, Spectro) is consistent with small-sample regimes where prior exposure to the same datasets can unduly benefit in-distribution baselines.

# I  SUPPLEMENTARY EVALUATION ON WOODS

To further substantiate the claim in Section 4.4 that CAUKER enables sample-efficient pre-training for zero-shot classification, we evaluate on the WOODS benchmark (Gagnon-Audet et al., 2023), which targets out-of-distribution (OOD) generalization in time-series tasks. We include 17 datasets from WOODS; notably, 12 of them (CAP, SEDFx, and MI families) are EEG-based, thereby probing a domain that is distributionally distant from the UCR suite used in our main evaluation. We compare three contenders:

1. **ERM (supervised baseline).** The carefully designed empirical risk minimization pipeline used in the original WOODS paper (Gagnon-Audet et al., 2023).

2. **Mantis-2M (real-data pre-training).** The original Mantis encoder (Feofanov et al., 2025) pre-trained on $\sim$1.89M real-world series.

3. **CauKer100K (synthetic pre-training).** A Mantis encoder pre-trained from scratch on only $100\,\mathrm{K}$ CAUKER samples (Section 3); all other components and evaluation protocol are kept identical to Mantis-2M.

Following our zero-shot protocol, we freeze the encoder and train a lightweight classifier on top of embeddings using only the in-benchmark training split of each WOODS dataset. Unless otherwise stated, we report standard classification accuracy (averaged over the official splits) and highlight the best result per (sub)domain in bold.

**Results**   Table 10 summarizes domain-level averages (aggregated over constituent datasets) and overall statistics. Despite using roughly $20\times$ fewer pre-training samples than Mantis-2M, CauKer100K attains the strongest average performance and secures 12 wins (including ties) out of 17 datasets. The advantage is pronounced on EEG-heavy families (CAP, SEDFx), while MI favors the supervised ERM baseline, suggesting that certain EEG sub-tasks may still benefit from label-rich supervised specialization. On HAR (human activity recognition), CAUKER again edges out both alternatives.

These results corroborate our central message: CAUKER pre-training yields robust and transferable representations that generalize beyond the UCR distribution, even to EEG-centric WOODS tasks. Crucially, this benefit is achieved with an order of magnitude fewer pre-training samples than the real-data corpus used by Mantis-2M, reinforcing the sample-efficiency of synthetic, causally structured time-series generation.

## J    PRE-TRAINING CHRONOS ON CAUKER AND ZERO-SHOT EVALUATION

To assess whether the proposed synthetic pipeline CAUKER also transfers to forecasting pre-training, we trained Chronos models from scratch on a purely synthetic corpus of

$$N = 1{,}000{,}000 \quad \text{univariate series of length} \quad L = 512,$$

which amounts to approximately $N \times L \approx 0.512\text{B}$ time points ("observations"). We abbreviate this setting as CauKer1M[5] in the results table for consistency with our internal logs. We evaluate in a zero-shot manner on the chronos-zero-shot benchmark, comprising 27 subsets explicitly curated to be disjoint from the official Chronos pre-training mixture.

**Metric Mean Absolute Scaled Error**    We report Mean Absolute Scaled Error (MASE), a scale-free metric where lower is better. For a seasonal period $m$ and forecast horizon $H$, with ground truth $\{y_t\}_{t=1}^{T+H}$ and predictions $\{\hat{y}_t\}_{t=T+1}^{T+H}$, MASE is

$$\text{MASE} \ = \ \frac{1}{H} \sum_{t=T+1}^{T+H} \frac{|y_t - \hat{y}_t|}{\frac{1}{T-m} \sum_{i=m+1}^{T} |y_i - y_{i-m}|}. \tag{1}$$

Under this normalization, the seasonal naive forecaster attains MASE $\approx 1$ by construction, providing a meaningful baseline across heterogeneous series.

**Results.**    Table 11 contrasts zero-shot MASE for official Chronos checkpoints (trained on an 84B-observation mixture) against models pre-trained only on CAUKER. Despite using an order of magnitude fewer observations and no real data, CAUKER pre-training yields competitive zero-shot accuracy across Chronos model scales, and substantially outperforms the Seasonal Naive baseline.

Table 11: Zero-shot forecasting on the chronos-zero-shot suite (27 non-overlapping subsets). Lower MASE is better. CauKer1M denotes pre-training on 1M sequences of length 512.

| Model Type | Training Data | pre-training Data Size | MASE |
|---|---|---|---|
| Chronos Tiny | Official | 84B Observations | 0.87 |
| | CauKer1M | 0.5B Observations | 0.89 |
| Chronos Mini | Official | 84B Observations | 0.84 |
| | CauKer1M | 0.5B Observations | 0.87 |
| Chronos Small | Official | 84B Observations | 0.83 |
| | CauKer1M | 0.5B Observations | 0.86 |
| Chronos Base | Official | 84B Observations | 0.81 |
| | CauKer1M | 0.5B Observations | 0.83 |
| Seasonal Naive | – | – | 1.0000 |

---

[5]The label CauKer1M follows our internal shorthand and denotes the run with 1M sequences ($\approx 0.512\text{B}$ observations).

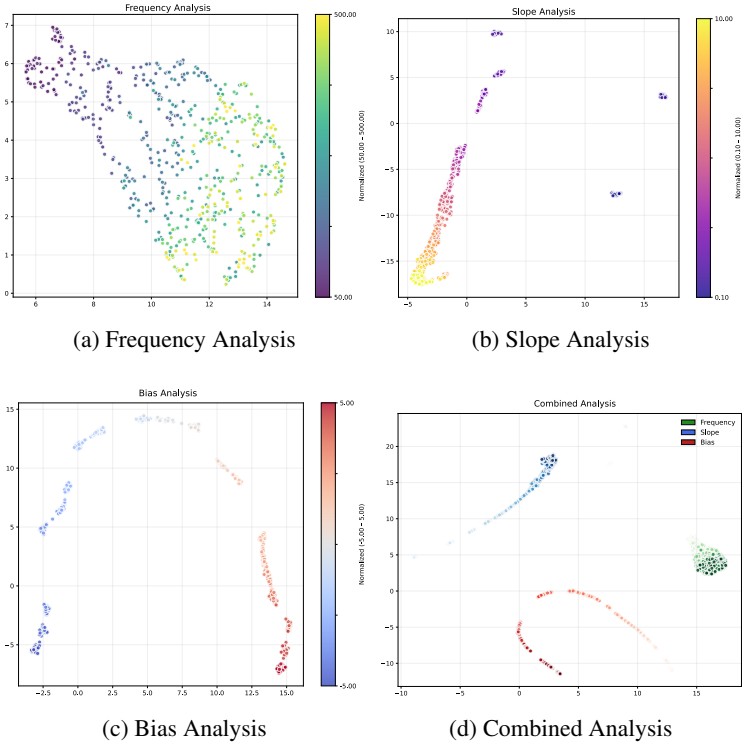

(a) Frequency Analysis

(b) Slope Analysis

(c) Bias Analysis

(d) Combined Analysis

Figure 11: UMAP projections of embeddings produced by the CauKer pre-trained encoder. Colour encodes the generating parameter for each synthetic class (green = frequency, blue = slope, red = bias).

**Discussion.** Despite pre-training on $\approx 160\times$ fewer observations ($\sim 0.5$B vs. 84B), the Chronos models trained on CAUKER lag the official checkpoints by only $\approx 2$–$3\%$ in zero-shot MASE. indicating that (i) CAUKER supplies sufficiently rich temporal structure for large models to leverage in a zero-shot regime, and (ii) a purely synthetic corpus—designed originally for classification—can still endow forecasting FMs with strong generalization, despite the absence of any real-world pre-training data. This aligns with our broader finding that principled synthetic design can act as an effective substitute for large, curated real datasets when coupled with appropriate inductive biases and scale.

## K  VISUALIZATION OF EMBEDDINGS

We generated univariate time series of length $L = 512$ using the CAUKER pipeline. For the frequency class, 20 periodic kernels with periods evenly spaced in $[50, 500]$ were used. For the slope class, we sampled slopes in $[0.1, 10.0]$, and for the bias class, biases were drawn from $[-5, 5]$. Each parameter setting was instantiated 30 times to ensure balanced coverage across the range. We use Mantis 8M trained on 10M CauKer data to encode the time series.

The UMAP projections reveal that the encoder learned structured and disentangled representations:

- In the frequency, slope, and bias views (Figures 11a–11c), we observe continuous colour gradients along one principal direction of the embedding, confirming that the encoder preserves the underlying generative factor in a smooth and ordered fashion.

- In the combined view (Figure 11d), embeddings from the three generation processes form distinct clusters with minimal overlap, indicating that the encoder effectively disentangles the semantic attributes of each synthetic category.

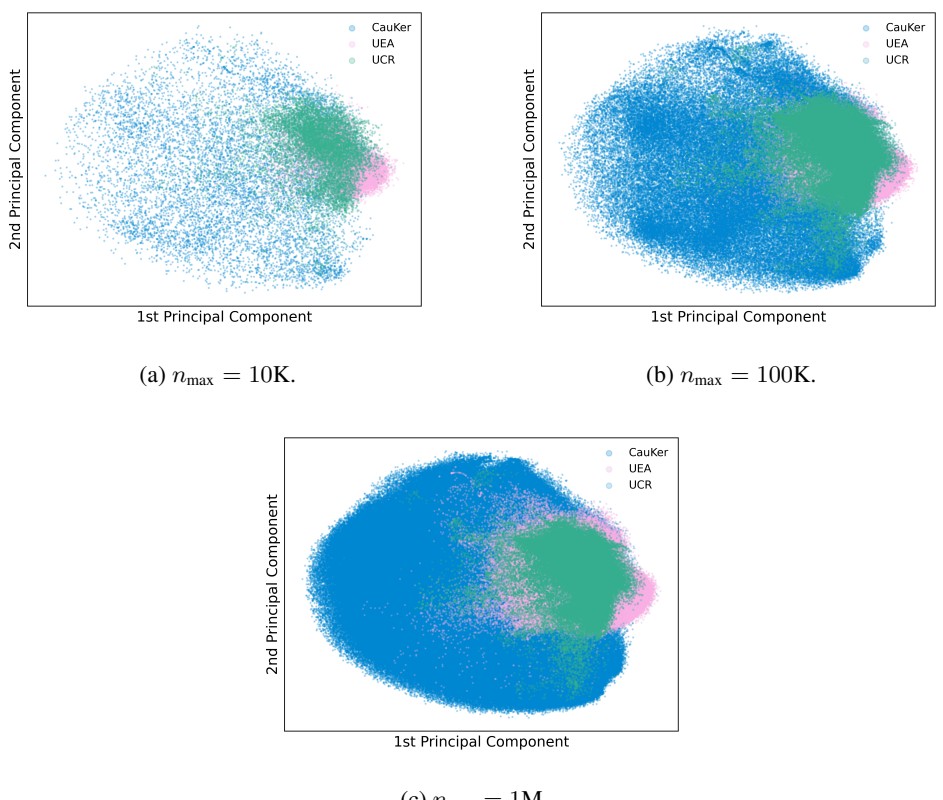

(a) $n_{\max} = 10$K.

(b) $n_{\max} = 100$K.

(c) $n_{\max} = 1$M.

Figure 12: PCA-visualization of Mantis embeddings for samples from UCR, UEA and CauKer-generated data. For each plot, we randomly select $\min(n_{\text{samples}}, n_{\max})$ samples for each dataset, where $n_{\text{samples}}$ is the dataset size and $n_{\max} \in \{10\text{K}, 100\text{K}, 1\text{M}\}$.

- The alignment of UMAP geometry with the known generative parameters supports the conclusion that the model did not merely memorize waveform patterns, but instead internalized semantically meaningful features of the data.

These results confirm that synthetic pre-training on CAUKER enables the encoder to learn robust, interpretable, and transferable representations even in the absence of real data.

Next, we evaluate the diversity of generated samples by comparing them to real benchmarks such as UCR and UEA. To this end, we visualize embeddings of time series samples using PCA projection onto the first two principal components. We use the original pre-trained Mantis as the encoder and compute the PCA on the concatenation of 1 million CauKer-generated samples, UCR and UEA data. The results are shown in Figure 12; for each plot, we matched the number of UEA and CauKer samples to enable a fair comparison of data distributions. As can be seen, CauKer generates more diverse samples, spanning the embedding space more uniformly. This broader coverage may facilitate pre-training by encouraging the learning of more generalizable features. This is consistent with our empirical findings: CauKer data achieves comparable performance to the original Mantis pre-training dataset (which includes UCR) on UCR (Table 9) and outperforms it on the WOODS benchmark (Table 10).

## L    ATTENTION ROLLOUT ANALYSIS.

To further compare representations learned from synthetic and real data, we apply the Attention Rollout (Abnar & Zuidema, 2020) to Mantis pre-trained on CAUKER-100K and to the original Mantis checkpoint trained on 1.89M real-world corpora. As illustrated in 13, for randomly selected

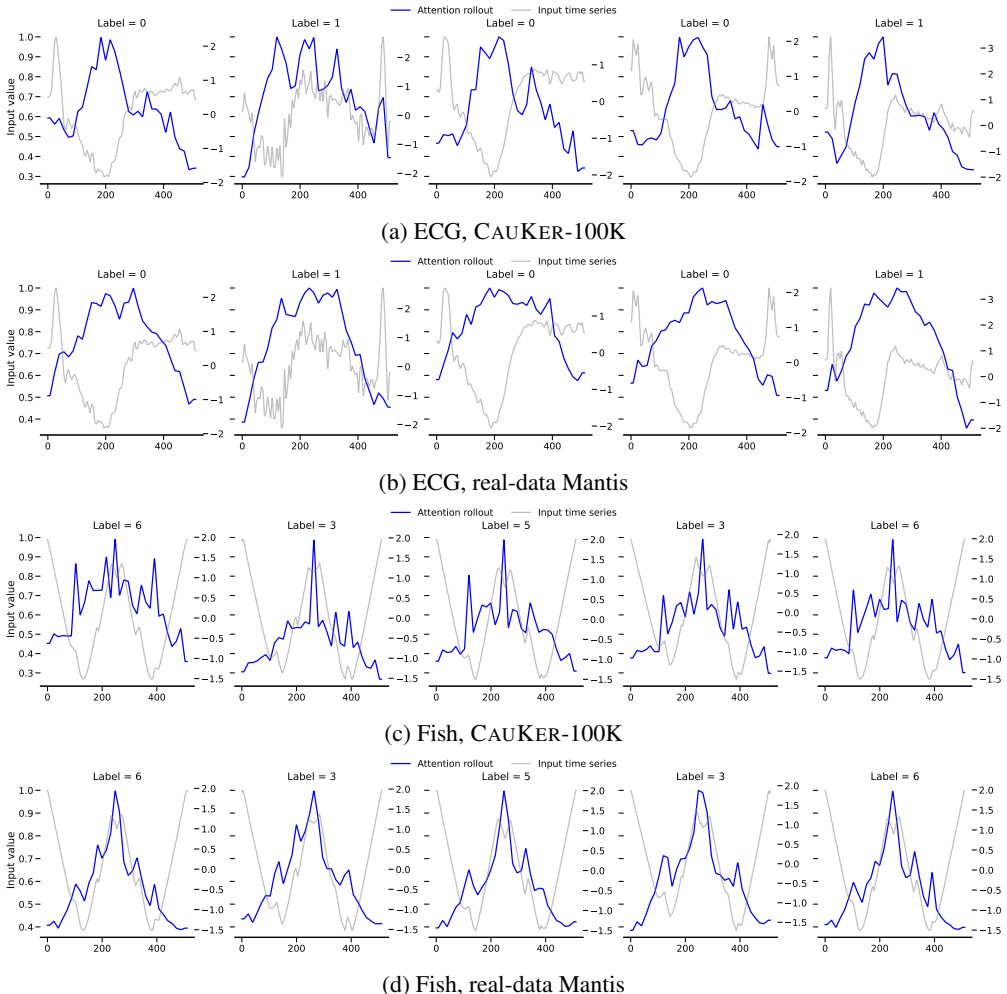

(a) ECG, CAUKER-100K

(b) ECG, real-data Mantis

(c) Fish, CAUKER-100K

(d) Fish, real-data Mantis

Figure 13: Attention Rollout on UCR ECG and FISH samples.

representative examples from the UCR ECG and UCR Fish datasets, the CAUKER pretrained model exhibits noticeably sharper and more localized attention maps: its aggregated attention mass concentrates on short subsequences containing visually salient, class-discriminative patterns, whereas the original Mantis tends to distribute attention more diffusely along the series. This suggests that the causal structure and diversity of CAUKER encourage TSFMs to focus more strongly on discriminative temporal segments.

## M ADDITIONAL EXPERIMENTS ON DOWNSTREAM FINE-TUNING

To complement the zero-shot evaluation in the main paper, we also study downstream fine-tuning of Mantis on UCR. Specifically, we follow the default fine-tuning pipeline provided in the original Mantis implementation (same optimizer, schedule, data splits, and classifier head), and compare three pre-training configurations: (i) the original Mantis model trained on the real-data corpus of 1.89M series, and (ii) two models pre-trained on CAUKER synthetic data with 100K and 1M series, respectively. Table 12 summarizes the resulting UCR test accuracies. We observe that all configurations benefit from supervised fine-tuning compared to their zero-shot counterparts (Table 7), and that increasing the size of the CAUKER corpus from 100K to 1M substantially narrows the performance gap to the original real-data model. This indicates that CAUKER-pretrained models not only provide strong zero-shot representations, but also serve as a competitive initialization for downstream supervised adaptation.

Table 12: Downstream fine-tuning accuracy on UCR for Mantis using the default fine-tuning pipeline. All models are pre-trained with the indicated corpus and then fine-tuned under identical settings.

| Pre-training corpus | # pre-train series | Test accuracy |
|---|---|---|
| Original Mantis real-data corpus | 1.89M | 0.8496 |
| CAUKER | 100K | 0.8291 |
| CAUKER | 1M | 0.8457 |

Table 13: Zero-shot-style evaluation on irregular, multivariate clinical benchmarks. We compare the original Mantis encoder with Mantis pre-trained on CAUKER-100K and CAUKER-1M.

| Dataset | Model | AUROC | AUPRC |
|---|---|---|---|
| P12 | Mantis (real-data) | 0.8121 | 0.4340 |
| | Mantis (CAUKER-100K) | 0.7984 | 0.4276 |
| | Mantis (CAUKER-1M) | 0.8189 | 0.4592 |
| P19 | Mantis (real-data) | 0.8846 | 0.5368 |
| | Mantis (CAUKER-100K) | 0.8534 | 0.4954 |
| | Mantis (CAUKER-1M) | 0.8709 | 0.5005 |

## N  SUPPLEMENTARY EVALUATION ON IRREGULAR TIME SERIES

To assess whether CAUKER pre-trained encoders generalize beyond regularly sampled, fixed-length benchmarks such as UCR and WOODS, we additionally evaluate Mantis on two irregular, multivariate clinical datasets, P12 (Goldberger et al., 2000) and P19 (Reyna et al., 2019). Both datasets consist of sparsely and irregularly sampled physiological measurements with highly imbalanced binary labels (Li et al., 2023a; Zhang et al., 2021).

We follow the same frozen-encoder, zero-shot-style protocol as in the main experiments: the Mantis encoder is either the original real data pre-trained checkpoint, or pre-trained from scratch on CAUKER with 100K or 1M synthetic series. As standard in this setting, we report AUROC and AUPRC on the held-out test split. The results in Table 13 show that CAUKER-generated data outperforms the original Mantis encoder on P12, whereas on P19 the CAUKER-1M model achieves an AUROC of 0.8709 and an AUPRC of 0.5005 compared to 0.8846 and 0.5368 for the real-data pretrained baseline.

## O  USE OF LARGE LANGUAGE MODELS

Large Language Models (LLMs) were used as a general-purpose writing assistance tool during the preparation of this manuscript. Specifically, LLMs were employed for language refinement to improve the clarity, grammar, and style of technical writing while preserving the original scientific content and authorial voice. The LLMs did not contribute to research ideation, experimental design, data analysis, or the formulation of scientific conclusions. All technical innovations, methodological contributions, experimental results, and scientific insights presented in this work are entirely the intellectual product of the human authors. The authors take full responsibility for all content, including any portions refined with LLM assistance, and have verified the accuracy and appropriateness of all information presented.

