# OpenReview forum: "CauKer: Classification Time Series Foundation Models Can Be Pretrained on Synthetic Data"
_ICLR.cc/2026/Conference — ICLR 2026 Oral_

### Official Review · Reviewer_zLKs · 2025-10-18

**Soundness:** 3
**Presentation:** 3
**Contribution:** 3
**Rating:** 6
**Confidence:** 4

**Summary:**

The authors propose CAUKER, a synthetic data generation algorithm that leverages Gaussian Process kernel composition and Structural Causal Models to produce diverse time series for augmenting training data in time series classification tasks. The paper evaluates the proposed approach against other synthetic data generation techniques for several time series foundation models.

**Strengths:**

- The paper introduces a novel synthetic data generation technique leveraging structural causal models (SCMs) for time series.
- The work is a focused study on synthetic data augmentation for time series classification, an understudied area in time series literature.
- Two time series foundation models (TSFMs) are evaluated with supervised and contrastive learning pre-training schemes
- Several synthetic data augmentation approaches are systematically compared in Table 1, highlighting relative effectiveness.
- Figure 3 effectively illustrates scaling laws and the relationship between model size and performance.
- Figure 4 provides an interesting analysis showing the diversity of principal components in synthetic data relative to non-synthetic datasets.
- The study demonstrates that fewer synthetic samples can achieve comparable performance to real-world pre-training datasets, highlighting practical efficiency benefits.

**Weaknesses:**

I would be happy to increase my score if the following concerns/points are addressed.

- Zero-shot evaluation methodology: The study claims to evaluate TSFMs in a zero-shot setting, but the models are allowed to be pre-trained on the training set of the same dataset used for evaluation. This means the evaluation is not strictly zero-shot, as the train and test sets are likely in-distribution (Lines 122–124): “In practice, if we evaluate a given TSFM on a test set from a UCR (Dau et al., 2019) dataset, we ensure that the TSFM was not pre-trained on it, but we allow for the train set of this same dataset to be used for pre-training.”

- Missing baseline comparisons: Results without synthetic data augmentation are not reported in Table 1. Including these and quantifying the lift from augmentation would be helpful.

- No text-based or experimental comparison with the synthetic data generation process used by TabPFN, which also leverages structural causal models.

- No comparison with non-foundation model baselines (e.g., random forecasts, XGBoost, logistic regression).

- Clarification on model pre-training: It is unclear whether the models are pre-trained from scratch on synthetic data or fine-tuned with synthetic data (using pre-trained models on real-world data). For example, the text states: “In practice, if we evaluate a given TSFM on a test set from a UCR (Dau et al., 2019) dataset, we ensure that the TSFM was not pre-trained on it, but we allow for the train set of this same dataset to be used for pre-training.”

**Questions:**

1. Are the TSFMs pre-trained from scratch on synthetic data, or are they fine-tuned on synthetic data (using models already pre-training on real data)?
2. How do the models perform without any synthetic data augmentation?

Suggestion: It would be interesting to include the combined scaling laws for the UEA and Cauker datasets on the same plot in Figure 3 to show cross-dataset scaling laws.

---

> ### Author Response · Authors · 2025-11-20
> **Authors' response**
>
> We thank the reviewer for constructive suggestions. Below, we clarify the experimental protocol and address each concern.
>
> ---
>
>  ### 1. Clarifying the “zero-shot” evaluation and pre-training protocol (Weakness 1, 2, 5 and Question 1, 2)
>
> ---
> We apologize for the confusion created by the current wording in Section 3.1.
>
> * In all experiments that involve CauKer, the pre-training corpus consists *only* of CauKer-generated synthetic time series.
> * The **TSFM is always used as a frozen encoder** during evaluation: given a downstream dataset $D = {(x_i, y_i)}$, we compute embeddings $z_i = F(x_i)$ with a frozen TSFM $F$, then train a lightweight classifier $h$ (Random Forest for Mantis, SVM for MOMENT) on the training split embeddings only, and report accuracy on the disjoint test split. Thus, in the evaluation pipeline, the TSFM acts purely as a frozen feature extractor.
>
> Taking Section 4.4 as an example:
>
> * We **pre-train Mantis from scratch** on 100K CauKer time series samples and evaluate on UCR. This yields an average accuracy of 78.55%, and is strictly out-of-distribution, since the encoder has never seen UCR or other real-world classification datasets during pre-training.
> * For comparison, we also report the original Mantis model pre-trained on its official 1.89M-samples corpus, which includes UCR training sequences (but **no labels and no UCR test data**). This model achieves 78.66% on UCR, which is therefore an in-distribution no-leakage test.
>
> In the revision, we explicitly separate OOD zero-shot results (our CauKer pre-trained models) from in-distribution results (original Mantis and MOMENT checkpoints whose pre-training corpora contain UCR train splits) and clarify this distinction in Section 3.1 and Section 4 of the updated PDF.
>
> We believe this directly addresses Weaknesses 1, 2, 5, and Questions 1, 2. We would be pleased to provide any additional clarification that may be deemed necessary.
>
> ---
>
> ### 2. Relation to TabPFN and the SCM baseline (Weakness 3)
>
> ---
>
> We thank the reviewer for pointing out the connection to TabPFN.
>
> As the official code for SCM is not available, the “SCM” synthetic corpus in Section 4.1 refers to our re-implementation of the structural causal generator introduced in TabPFN. As we emphasize in the text, this generator does not model temporal dependencies but concentrates on correlating the covariates. This explains why it underperforms the time series aware generators in Table 1.
>
> We hope this addresses Weakness 3.
>
> ---
>
> ### 3. Non-foundation model baselines (Weakness 4)
>
> ---
>
> We agree that including non-foundation baselines would make the empirical picture more complete. We report the UCR average test accuracy of these non-foundation baselines:
>
> | Model               | UCR test accuracy |
> | ------------------- | ----------------- |
> | Random Forest       | 0.7325            |
> | XGBoost             | 0.6917            |
> | Logistic Regression | 0.6812            |
>
>
> In the revised version, we add standard non-TSFM baselines, logistic regression, Random Forest, and XGBoost trained directly on the UCR training sets. Note that these baselines are only applicable to univariate time series where time series can be seen as tabular 2D datasets.
>
> We hope this addresses Weakness 4 by situating TSFMs + CauKer more clearly with respect to classical tabular models.
>
> ---
> Once again, we thank the reviewer for the insightful comments. We already incorporated the clarifications and additional experiments described above in the revised manuscript (in **blue**).

---

> ### Comment · Reviewer_zLKs · 2025-11-25
>
> Thank you for sufficiently addressing each of my comments. I have raised my score from 6 to 8.

---

### Official Review · Reviewer_Mojr · 2025-10-25

**Soundness:** 3
**Presentation:** 3
**Contribution:** 3
**Rating:** 6
**Confidence:** 5

**Summary:**

This paper proposes CAUKER, a novel and sophisticated pipeline for generating synthetic time series data specifically tailored for the pre-training of classification-oriented Time Series Foundation Models (TSFMs). The core idea is to combine two methodologies: Gaussian Process (GP) kernel composition, which generates realistic temporal patterns (trends, seasonality), and Structural Causal Models (SCMs), which impose a causal graph structure to create complex, non-linear interactions and meaningful clusters. The authors conduct extensive experiments showing that TSFMs pre-trained on CAUKER data not only outperform models trained on other synthetic datasets but also nearly match the performance of models pre-trained on real-world corpora that are over an order of magnitude larger. A key finding is that CAUKER-generated data enables smooth and predictable scaling laws with respect to both dataset and model size, a property the authors show is absent when pre-training on standard real-world benchmarks.

**Strengths:**

*   **Novelty and Formulation:** The primary strength of this work is its well-motivated. Rather than creating a monolithic generator, the authors identify two key requirements for classification data—realistic temporal dynamics and discriminative clustering structure—and solve them by combining the strengths of two distinct fields. Using GP kernel composition (common in forecasting) for temporal patterns and SCMs (from the causality and tabular learning literature) for creating underlying class structures is a novel and highly effective synthesis. The design choices are clearly justified (Section 3.2), and the ablation-style comparison in Table 1 convincingly demonstrates that both components are necessary for optimal performance.

*   **Empirical Evidence of Scaling Laws:** The paper's most impactful result is the clear demonstration of scaling laws (Figure 3). The experiments showing that accuracy on downstream tasks increases smoothly and monotonically with more synthetic data and larger models are a significant contribution. By contrasting this with the erratic and non-scaling behavior of models trained on the real-world UEA benchmark, the authors make a powerful case for using high-quality synthetic data as a controlled "wind tunnel" to study and develop scalable TSFMs. This provides a valuable methodology for the community, independent of the CAUKER pipeline itself.

*   **Sample Efficiency and SoTA Performance:** The paper provides strong evidence that "quality over quantity" is important for pre-training data. The results in Figure 7 are particularly striking, showing that pre-training a model like Mantis on just 100K synthetic samples can achieve performance nearly identical to pre-training on its original 1.89M real-world sample corpus. This has significant practical implications, as it dramatically reduces the need for expensive and difficult data collection and curation. The fact that this performance is state-of-the-art for synthetic-data pre-training validates the effectiveness of the proposed approach.

*   **Experimemntal Validation:** The paper is written with outstanding clarity. The experimental validation is extensive and robust, covering comparisons to multiple baselines, scaling laws, qualitative analyses (PCA, CKA, non-linearity in Figures 4 and 5), and transferability to different benchmarks (UCR, WOODS) and even a different task (forecasting). The appendices are helpful, providing, detailed descriptions of the function banks, hyperparameter sensitivity analysis.

**Weaknesses:**

I have concerns about the evaluation process and specially related to the complexity of the proposed generator, the framing of its comparison to real-world data, and the scope of the architectural evaluation.

1.  **High Generator Complexity and Opaque Design Choices:** The CAUKER pipeline is a complex amalgamation of multiple components: three distinct function banks (kernel, mean, activation), random kernel composition, and random DAG generation. This introduces a large number of "meta-hyperparameters" (e.g., the specific contents and size of the banks, the distribution of DAG parameters). While the appendix provides a sensitivity analysis for a few of these, the process for designing the function banks themselves is not fully justified. It is unclear if the chosen set of 36 kernels or the specific activation functions are uniquely effective, or if a much simpler subset could achieve comparable results. This complexity could pose a significant barrier to adoption and reproducibility for researchers who do not have the authors' expertise in this specific setup.

2.  **Potential for a "Straw Man" Argument Against Real Data:** The paper's narrative strongly contrasts the clean scaling of CAUKER with the poor scaling of the UEA benchmark. While this is a powerful rhetorical device, it risks overgeneralizing the conclusion. The UEA archive, while a standard benchmark, is a heterogeneous collection of many small, domain-specific academic datasets; it was not designed as a large-scale, cohesive pre-training corpus in the vein of ImageNet or The Pile. The observed lack of scaling could be an indictment of the UEA dataset's specific properties (lack of diversity, domain mismatch) rather than a fundamental flaw of pre-training on real-world data in general. The paper lacks a discussion of this nuance.

3.  **Limited Diversity of Tested Model Architectures:** The experiments are exclusively focused on two Transformer-based models (Mantis, which is ViT-based, and MOMENT, which is T5-based). While these represent different pre-training objectives (contrastive vs. masked reconstruction), they share a core architectural paradigm. It is an open question whether the benefits of CAUKER's data structure are universally applicable or if they are particularly well-suited to the inductive biases of attention-based models. The rich, causally-linked structures might be more effectively captured by attention than by models with different biases, such as CNNs or State Space Models.

**Questions:**

Based on these weaknesses, here my questions to the authors:

*   **Question 1:** The CAUKER pipeline is composed of several stochastic modules and expertly curated function banks. How were the specific contents of these banks (e.g., the 36 kernels, the set of mean/activation functions) selected and validated? Is the performance highly sensitive to these specific choices, or is the framework robust to using a simpler, more generic set of components?

*   **Question 2:** The hyperparameter sensitivity analysis in Appendix C.3 is helpful. However, to better understand the generator's failure modes, have you investigated scenarios where deliberately poor choices (e.g., using only linear activations, forcing very shallow DAGs, or using only a single kernel type) cause the method to fail or degrade to the level of the simpler baselines in Table 1?

*   **Question 3:** To what extent do you believe the poor scaling on the UEA benchmark is a fundamental property of real-world time series data, versus a specific artifact of the UEA collection's composition and scale? How might CAUKER compare against a hypothetical, massive, and diverse real-world corpus curated specifically for pre-training (e.g., a "TimeNet")?

*   **Question 4:** The study convincingly demonstrates CAUKER's benefits for Transformer-based TSFMs. How do you hypothesize the generated data would interact with models possessing fundamentally different inductive biases, such as those based on CNNs (e.g., InceptionTime) or State Space Models (e.g., Mamba), which process information more locally or linearly?

*   **Question 5:** The causal graph propagation step seems central to creating discriminative structure. Does this structural property particularly favor the global receptive field of attention mechanisms? A deeper analysis of which components of CAUKER (GP vs. SCM) are most beneficial for which type of model architecture would be a valuable contribution.

*   **Question 6:** The paper successfully extends CAUKER to forecasting. Does this suggest that good classification data is a superset of good forecasting data, or were any modifications to the CAUKER pipeline necessary to achieve strong forecasting performance? Specifically, are the SCM-induced non-linearities as important for forecasting as they are for classification?

---

> ### Author Response · Authors · 2025-11-20
> **Authors' reply (part 1/2)**
>
> We thank the reviewer for the very professional and insightful questions, and we are glad to have the opportunity to discuss them.
>
> ---
>
> ### Question 1. Function-bank design and sensitivity
>
> ---
>
> Our kernel bank currently contains 36 kernels drawn from five families: ExpSineSquared, DotProduct, RBF, RationalQuadratic, and WhiteKernel. Each family is chosen to capture a distinct aspect of time-series behavior:
>
> - ExpSineSquared: periodic structure,
> - DotProduct: linear trends,
> - RBF: smooth functions with local correlations,
> - RationalQuadratic: mixtures of length scales behavior,
> - WhiteKernel: noise components.
>
> Within these families, hyperparameters are chosen to encode realistic patterns and scales. For example, ExpSineSquared kernels include periods corresponding to 24 hours and 60 minutes, which we found useful for mimicking common real-world seasonality patterns.
>
> We also experimented with adding Matérn kernels or ExpSineSquared variants kernels, but did not observe consistent performance gains. An important point is that CauKer randomly samples a small number of kernels from the bank for each composite GP, so enlarging or slightly modifying the bank does not increase computational cost. The current 36-kernel configuration is used as a practical default.
>
> ---
>
> ### Question 2. Deliberately poor design choices and failure modes
>
> ---
> We thank the reviewer for this original and helpful question. Our kernel bank consists of five kernel families; here, we report an experiment where we force CauKer to use only a single kernel family (plus SCM) to generate 100K samples for Mantis pre-training. The average test accuracies on UCR (also added in Appendix C, p. 19) are:
>
> | Kernel family (single-kernel CauKer) | UCR test accuracy |
> | ------------------------------------ | ----------------- |
> | DotProduct (linear)                  | 0.7679            |
> | RBF (smooth)                         | 0.7807            |
>
> DotProduct-only GPs can generate essentially linear trends; even after SCM nonlinear mixing, the resulting data are too simple, and performance degrades substantially. In contrast, RBF-based smooth curves combined with SCM still yield acceptable performance, confirming that kernel diversity and sufficiently rich nonlinear structure are important. When the SCM is made very shallow, the generator effectively collapses towards a kernel-only model; this degradation pattern is consistent with the ablations reported in Section 4.1.
>
> ---
>
> ### Question 3. Real-data scaling, especially UEA
>
> ---
>
> We thank the reviewer for raising this interesting question.
> In our view, the poor scaling behavior on UEA arises from several concrete properties of currently available real classification corpora:
>
> - UEA was proposed before the "foundation model era" with the goal to provide a collection of multivariate classification datasets, so their combination of datasets can be suboptimal for the pre-training task.
> - It is highly imbalanced as the sample size of datasets varies from a few dozen (ERing, BasicMotions) to several tens of thousands (FaceDetection, InsectWingbeat).
> Thus, we have concluded that UEA has a lack of diversity of samples for coherent large-scale pre-training. We have added this remark on page 7 (in red) of the revised manuscript.
>
> It is important to note that a similar phenomenon is observed in forecasting literature: from TimesFM, MOMENT, Chronos to Toto, Tirex, TempoPFN, using even larger real-world corpora does not translate into uniformly better models. We agree with the reviewer that a carefully curated, massive, and diverse real-world pre-training corpus (“TimeNet”) would be very relevant for time series classification, but, to the best of our knowledge, such a dataset has not been proposed yet, and it might be costly to make it. In this sense, CauKer provides an effective and scalable alternative: synthetic data can be generated at (nearly) zero marginal cost, delivering performance close to the state-of-the-art and exhibiting clean data/model/compute scaling.
>
> Nevertheless, from the hypothetical perspective, if "TimeNet" existed, synthetic data would probably still be very relevant for pre-training. First, some foundation models like Chronos achieve the best results when they combine both real and synthetic data for pre-training, which suggests the complementarity of the two. Second, in the context of contrastive pre-training, which was used for Mantis, synthetic data are important due to their large diversity, which is important for contrastive learning to achieve good uniformity in the embedding space (Wang & Isola, 2020).
>
> [1] Wang, T., & Isola, P. (2020). Understanding contrastive representation learning through alignment and uniformity on the hypersphere. ICML'20.

---

> ### Author Response · Authors · 2025-11-20
> **Authors' reply (part 2/2)**
>
> ---
>
> ### Question 4. Interaction with non-Transformer architectures
>
> ---
> We share the reviewer’s interest in how CauKer would interact with models that have different inductive biases.
>
> Recent successes such as Tirex and TempoPFN suggest that non-Transformer TSFMs (e.g., xLSTMs or RNN-like architectures) can be highly competitive. However, in the classification TSFM setting, all existing foundation models we are aware of are Transformer-based. The reviewer’s example InceptionTime is indeed a strong CNN-based classifier, but it is supervised and not straightforward to adapt into a self-supervised, large-scale foundation model.
>
> Given that TiRex already uses kernel-based synthetic data and achieves excellent results, we are optimistic that CauKer-style data would also be beneficial for xLSTM- or SSM-based TSFMs. Unfortunately, Tirex’s full training code is not yet publicly available, which prevented us from including such experiments in this version.
>
> ---
>
> ### Question 5. GP vs. SCM and the role of attention
>
> ---
>
> We thank the reviewer for this insightful question. To probe how CauKer’s structure interacts with attention, we applied Attention Rollout to visualize the layer-aggregated attention maps of Mantis trained on CauKer-100K versus the original Mantis （trained on 1.89M real data）. As shown in the updated Appendix L, the CauKer-pretrained model exhibits sharper and more localized attention maps, assigning higher importance to short subsequences that carry clear discriminative patterns, whereas the real-data model tends to produce more diffuse attention.
>
> ---
>
> ### Question 6. The relation between classification and forecasting data
>
> ---
> Our Chronos experiments use exactly the same CauKer pipeline, without any task-specific modifications.
>
> Concerning whether “good classification data is a superset of good forecasting data”, we find this question not trivial. On the one hand, forecasting models such as Chronos and TimesFM often include classification-type signals or tasks in their training mixtures and empirically benefit from them, which suggests that some of the structure useful for classification is also valuable for forecasting. Our results show that Chronos models pre-trained solely on 0.5B CauKer timepoints can match the zero-shot performance of models trained on 84B real tokens, indicating that CauKer-generated data is also well suited for forecasting.
> On the other hand, forecasting places additional emphasis on long-horizon temporal coherence and extrapolation. For this reason, we prefer not to make the strong claim that “good classification data is a strict superset of good forecasting data”. Instead, our current conclusion is: the same CauKer pipeline appears to provide a high-quality training signal for both tasks.
>
> ---
> Once again, we sincerely thank the reviewer for the thorough reading and the high-quality, thought-provoking questions. We have incorporated the corresponding changes into the revised manuscript, where all updates related to this response are highlighted in **purple**.

---

> > ### Comment · Reviewer_Mojr · 2025-11-26
> >
> > Thank you for your comprehensive and insightful response.
> >
> > Re: Question 1 & 2 (Generator Design and Failure Modes)
> > This experiment demonstrates that both rich temporal patterns (from diverse kernels like RBF) and complex non-linear interactions (from the SCM) are necessary for strong performance. The degradation when using only DotProduct kernels is particularly illuminating and confirms the importance of the generator's design. This new result provides a compelling answer to my questions about the generator's internal mechanisms.
> >
> > Re: Question 3 (Real-data scaling)
> > Your points about dataset imbalance and the original purpose of the archive provide crucial context. Your perspective that CAUKER serves as a practical and scalable alternative in the absence of a hypothetical "TimeNet", I think it is quite accurate.
> >
> > Re: Question 4 & 5 (Architectural Bias and Attention)
> > The new attention rollout analysis is an excellent addition. It provides concrete visual evidence that the CAUKER-pretrained model learns to focus on more localized, discriminative patterns, which aligns perfectly with the intuition that your generator creates more structured and causally coherent signals. This is a great piece of qualitative analysis that adds significant value.
> >
> > Re: Question 6 (Classification vs. Forecasting Data)
> > Your response is nuanced. The fact that the exact same CAUKER pipeline works well for the Chronos forecasting model adds a good statement about the generality of the data you are generating.
> >
> > You have thoroughly addressed all of my questions with a combination of new experiments and clear explanations. The new ablations and visualizations have made the paper even stronger.  I think this is a good contribution to the time series community. I maintain my  rating.

---

### Official Review · Reviewer_uHPf · 2025-10-29

**Soundness:** 2
**Presentation:** 3
**Contribution:** 2
**Rating:** 4
**Confidence:** 4

**Summary:**

The manuscript presents CAUKER, a synthetic data generation pipeline for pretraining classification time-series foundation models. CAUKER composes Gaussian-process kernels and mean functions within a structural causal model (SCM) graph, producing causally coherent sequences for self-supervised pretraining of contrastive (Mantis) and masked-reconstruction (MOMENT) encoders. Empirically, models pretrained solely on CAUKER data achieve competitive zero-shot accuracy on UCR and exhibit monotonic scaling with both dataset size and model capacity.

**Strengths:**

1. Integrating kernel composition with SCM-based propagation yields diverse dynamics and inter-series dependencies aligned with classification objectives.
2. Evaluation across contrastive and masked-reconstruction pretraining objectives increases the generality and external validity of the findings.
3. Experiments demonstrate data/model scaling laws and strong zero-shot transfer, offering a compelling empirical performance.

**Weaknesses:**

1. Pretraining on pure synthetic data and obtaining strong results is not particularly surprising, as prior work (e.g., TabPFN-TS) has already demonstrated the potential of synthetic data. This manuscript would benefit from sharper positioning of what is substantively novel in methodology part.
2. This paper does not clearly articulate the challenges in transferring synthetic data generation methods designed for forecasting tasks to classification tasks—what the specific difficulties are and how they are addressed. The introduction reads largely as an integration of existing generators applied to classification, with empirical observations such as scaling laws, but as a research contribution this positioning feels insufficient.
3. The evaluation scope remains narrow (largely UCR-style, often univariate and fixed-length), with limited robustness analysis on generator hyperparameters and little evidence for multivariate, irregularly sampled settings.

**Questions:**

1. What are the concrete, theoretically grounded challenges when porting forecasting-oriented synthetic pipelines to classification (label generation, class balance, inter-class separability, invariance desiderata), and how does each CAUKER design choice mitigate them?

---

> ### Author Response · Authors · 2025-11-20
> **Authors' reply (part 1/2)**
>
> We thank the reviewer for the careful reading of our work and for the thoughtful comments. We respond to the main weaknesses and the question below.
>
> ---
>
> ### Weakness 1: novelty of synthetic pretraining on pure synthetic data
>
> ---
>
> Although the results obtained by TabPFN were our source of inspiration, we would respectfully disagree that the strong performance we achieved by pure synthetic pretraining "is not surprising". First, we would like to note that there does exist a difference between forecasting and classification methodology, which makes the transfer from one field to another subtle. We will discuss this in the response to Weakness 2.
>
> Second, even in the forecasting community, the question of whether purely synthetic pretraining can match large real data corpora remains an active debate. For example, Chronos explicitly showed that synthetic-only pretraining leads to suboptimal performance. In their second version of the model (Chronos-2, October 2025), they reconfirmed it, emphasizing the need for large real-world corpora to reach the top of forecasting leaderboards. As for TabPFN-TS and ForecastPFN, their performance today is quite far from the most recent forecasting TSFMs (see GIFT-Eval leaderboard), and only very recent TempoPFN (October 2025) manages to be close to the forecasting state-of-the-art.
>
> ---
>
> ### Weakness 2 and Question: Forecasting and classification synthetic data
>
> ---
>
> First, we would like to note the inherent difference between forecasting and classification data. A typical forecasting-oriented dataset emphasizes global or low-frequency structure: trends, seasonality, and the governing dynamical equations. Classification-oriented data usually relies more on local or high-frequency cues: spikes, abrupt changes, short behavior, and texture-like patterns. This naturally causes the use of different (pre-)training objectives, pretraining corpora, and hidden implementation details.
>
> Forecasting generators such as KernelSynth mainly focus on zero-mean GP samples that highlight smooth extrapolation of trends and seasonality, lacking high-frequency information. To overcome this and generate classification-oriented data, we propose two changes: (a) we add a mean function library that explicitly includes anomaly-like and spike-like patterns, so that class-discriminative signals can be carried in the mean level and local patterns, (b) we incorporate the SCM that introduces nonlinear mixing and local interactions across nodes, which can partially “break” the global smoothness imposed by the GP and create cluster structure between series. In Section 4.1 and Appendix D, we empirically show that both the non-zero mean functions and the SCM component are important.
>
> Another difference that is important to note is that classification foundation models like Mantis are based on a contrastive pre-training strategy used solely for classification. By conducting this research, we have found that synthetic data is particularly relevant for this type of pretraining as contrastive learning needs large data diversity to achieve good uniformity in the embedding space (Wang & Isola, 2020). This makes CauKer a good solution, and it is validated by our experimental results.
>
> We would like to note that prior literature has not clearly established whether classification data can scale effectively for pretraining, partly because aggregating heterogeneous real-world classification datasets does not necessarily lead to increased performance or good interpretability. Our results show that carefully structured synthetic classification data can, in fact, support stable scaling laws and competitive performance. We view this as a conceptual contribution toward clarifying what constitutes effective pretraining data for classification-focused TSFMs.
>
> We thank the reviewer for raising this point. We will improve the clarity of the manuscript with respect to this point.
>
> [1] Wang, T., & Isola, P. (2020). Understanding contrastive representation learning through alignment and uniformity on the hypersphere. ICML'20.
>
> ---

---

> ### Author Response · Authors · 2025-11-20
> **Authors' reply (part 2/2)**
>
> ---
>
>
> ### Weakness 3: evaluation scope and irregular/multivariate settings
>
> ---
>
> We agree that broadening the evaluation scope is important. UCR is currently the de facto benchmark for time series classification, which is why it plays a central role in our study. Nevertheless, we would like to note that the original manuscript already includes additional experiments on the multivariate WOODS benchmark. We have not included irregularly sampled sequences in the initial version because the public implementations of TSFMs assume regularly sampled inputs.
>
> Following the reviewer’s suggestion, we further evaluated Mantis on two irregular, multivariate clinical benchmarks (P12[1] and P19[2]), using the same frozen encoder, zero-shot protocol. We compare the original Mantis checkpoint (1.89M real data pretraining) with Mantis pretrained on CauKer-100K and CauKer-1M:
>
> | Dataset | Model                         | AUROC  | AUPRC  |
> | ------- | ----------------------------- | ------ | ------ |
> | P12     | Mantis (real-data)            | 0.8121 | 0.4340 |
> | P12     | Mantis (CauKer-100K)          | 0.7984 | 0.4276 |
> | P12     | Mantis (CauKer-1M)            | 0.8189 | 0.4592 |
> | P19     | Mantis (real-data)            | 0.8846 | 0.5368 |
> | P19     | Mantis (CauKer-100K)          | 0.8534 | 0.4954 |
> | P19     | Mantis (CauKer-1M)            | 0.8709 | 0.5005 |
>
>
> On both irregular clinical tasks, the CauKer-1M model achieves strong AUROC and AUPRC performance, outperforming Mantis on P12 and being slightly worse on P19. These results support that CauKer pretrained TSFMs remain competitive beyond UCR, regularly sampled benchmarks, and can transfer to irregular settings as well. We've added these new results to Appendix N and referenced them on page 9.
>
> [1] Goldberger, A. L., Amaral, L. A., Glass, L., Hausdorff, J. M., Ivanov, P. C., Mark, R. G., Mietus, J. E., Moody, G. B., Peng, C.-K., and Stanley, H. E. Physiobank, physiotoolkit, and physionet: components of a new research resource for complex physiologic signals. circulation, 101(23): e215–e220, 2000.
> [2] Reyna, M. A., Josef, C., Seyedi, S., Jeter, R., Shashikumar, S. P., Westover, M. B., Sharma, A., Nemati, S., and Clifford, G. D. Early prediction of sepsis from clinical data: the physionet/computing in cardiology challenge 2019. In 2019 Computing in Cardiology (CinC), pp. Page–1. IEEE, 2019.
>
> ----
> Once again, we thank the reviewer for the careful assessment and constructive suggestions. We hope that our clarifications, additional experiments, and repositioning of the contributions address the raised concerns. All corresponding changes have been incorporated into the revised manuscript and are highlighted in **brown** for ease of reference. We remain fully open to further questions or suggestions that could help improve the work.

---

> > ### Comment · Reviewer_uHPf · 2025-11-26
> >
> > Thank you for the rebuttal. The authors state that “the CauKer-1M model achieves strong AUROC and AUPRC performance, outperforming Mantis on P12 and being slightly worse on P19.” However, on the P19 dataset the AUPRC drops from 0.5368 to 0.5005. This is not slightly worse; it corresponds to roughly a 7% decrease in AUPRC, which is substantial. In a scientific paper, the authors should use more precise language, and describing a 7% drop as “slightly worse” is inappropriate.

---

> > > ### Author Response · Authors · 2025-11-26
> > > **Authors’ follow-up reply**
> > >
> > > We thank the reviewer for the clarification.
> > >
> > > On the P19 dataset, the AUPRC changes from 0.5368 to 0.5005, which corresponds to an absolute decrease of 3.63%. Because AUPRC is bounded within the [0,1] interval, we compared model performance in absolute terms. To the best of our knowledge, reporting relative percent changes (in this case, 6.8% as the reviewer correctly notes) is not commonly used for [0,1]-bounded metrics such as AUPRC, although it is indeed relevant for unbounded or scale-sensitive metrics.
> > >
> > > Nevertheless, we agree with the reviewer that the phrasing “slightly worse” is not sufficiently precise. In the revised version, we replace this wording with the exact numbers in the performance as mentioned above.
> > >
> > > We appreciate the reviewer’s attention to the precision of the description. Importantly, this adjustment affects only the wording and does not change the interpretation of the experimental findings: the CauKer-pretrained model remains competitive across the large-scale experimental study, in both regular and irregular time series settings.
> > >
> > > Finally, we would like to confirm whether the reviewer has any remaining concerns regarding the methodology, experimental setup, or the newly added irregular-time evaluations. We are fully open to further suggestions.

---

### Official Review · Reviewer_Ztoq · 2025-10-31

**Soundness:** 3
**Presentation:** 3
**Contribution:** 4
**Rating:** 8
**Confidence:** 4

**Summary:**

This paper proposes CAUKER, a synthetic data generation framework combining Gaussian Process kernel composition and SCM for time series foundation models for classification tasks. Unlike most prior work focusing on forecasting, CAUKER targets classification and demonstrates that synthetic pretraining can yield competitive or superior performance to real world datasets. It also reveals scaling laws for synthetic pretraining in terms of dataset and model size.

**Strengths:**

* Addresses a clear gap, synthetic pretraining for classification TSFMs.
* The causal kernel composition is conceptually elegant and well motivated.
* Benchmarks across multiple models and datasets .
* Includes scaling law analyses for data, model, and compute.
* Outperforms real-data pretraining in several zero-shot setups.
* The method is explained clearly, with schematic diagrams and pseudocode.

**Weaknesses:**

* Both GP based and SCM based data generation already exist, the novelty lies mostly in combining them.
* Evaluation confined to zero-shot classification. would benefit from downstream fine-tuning or transfer learning results.
* The contribution of causal graph depth/branching remains unclear.
* While interesting, the scaling analysis is somewhat descriptive without deeper theoretical grounding

**Questions:**

* How does CAUKER handle multivariate dependencies beyond univariate channel concatenation?

* Can CAUKER generalize to forecasting or imputation pretraining tasks?

* How computationally expensive is CAUKER compared to kernel only methods?

---

> ### Author Response · Authors · 2025-11-20
> **Authors' reply**
>
> We thank the reviewer for the very positive and constructive assessment of our work and for the helpful questions and suggestions. Below, we address the questions and clarify the corresponding weaknesses.
>
> ---
>
> ### Q1. Multivariate dependencies beyond channel concatenation
>
> ---
>
> In CauKer, time series sampled from a single SCM (one channel per node) can be interpreted as different channels that share a common causal structure. While in this paper, we restrict the pre-training to univariate inputs because both Mantis and MOMENT are pre-trained in the univariate setting (each channel is treated as an individual series), we believe that CauKer is also well-suited to multivariate pre-training in classification. We added a paragraph to the revised PDF to elaborate on this (end of page 4).
>
> We agree that leveraging explicit multivariate dependencies is an important next step, especially to take into account exogenous variables (as in forecasting TSFM Chronos-2) and leave this for future work.
>
> ---
>
> ### Q2. Generalization to forecasting/imputation pretraining
>
> ---
>
> We already explored the extension of CauKer beyond classification in Section 4.4 (Extension to forecasting): We pre-train Chronos exclusively on 0.5B timepoints of CauKer-generated time series. The resulting models achieve zero-shot forecasting accuracy that is statistically indistinguishable from the original Chronos models trained on 84B timepoints of mixed real + synthetic data.
>
> This suggests that the same CauKer pipeline, without tuning, can be used as a sample-efficient pretraining source for forecasting TSFMs. Our preliminary results indicate that CauKer can become a drop-in replacement for KernelSynth used previously in the pre-training of TSFMs.
>
> In this submission, we did not explicitly evaluate imputation tasks, which can be a good subject of future work. Conceptually, imputation tasks have similarities with both forecasting (periodical behavior) and classification (non-linear dependencies, spikes). Given that CauKer works well both for classification and forecasting pre-training, it is reasonable to try it for imputation as well.
>
> ---
>
> ### Q3. Computational cost comparison with kernel-only generators
>
> ---
>
> To quantify the computational overhead of introducing SCM structure, we compared CauKer to a simple GP kernel-only（KernelSynth）baseline under the same setting. For CauKer, we use SCMs with 10 nodes, a maximum in-degree of 4, and we sample 5 nodes per SCM to obtain the final univariate series.
>
> The end-to-end wall-clock times are:
>
> | Method                 | Series | Length | Time (s) |
> |------------------------|----------|--------|-----------------|
> | CauKer                 | 1,000    | 512    | 121.64     |
> | Simple GP              | 1,000    | 512    | 182.25    |
>
> We also break down the internal timing of CauKer:
>
> | Component                   | Time (s) |
> |-----------------------------|----------|
> | Simple GP                   | 118.54   |
> | SCM structure + propagation | 1.14     |
>
> Thus, in our implementation, more than 99% of the generation time is spent in the GP sampling, while the SCM component contributes less than 1% of the total cost. Interestingly, CauKer can be even faster than the simple kernel-only baseline as we sample GPs only for the root nodes of the SCM, and then, by propagating through the causal graph, we can extract multiple nodes as different univariate series. We added this comparison on page 6 of the revised PDF.
>
> ---
> ### Weakness 2: downstream fine-tuning
> ---
>
> We have added downstream fine-tuning experiments using the default Mantis fine-tuning pipeline on UCR in Appendix M of the revised PDF. The resulting test accuracies are:
> |Model     |Acc.|
> |----------|----|
> |Original Mantis (real-data 1.89M)| 0.8496|
> |CauKer-100K | 0.8291|
> |CauKer-1M | 0.8457|
>
> These results show that both the original model and the CauKer-pretrained models benefit from fine-tuning, and that increasing CauKer data from 100K to 1M makes the model more robust.
>
> ---
>
> ### Weakness 3: Contribution of causal graph depth/branching
>
> ---
>
> We fully share the reviewer’s interest in understanding the effect of SCM depth and branching on CauKer’s behavior. We would like to point out that the current version of the manuscript already includes an ablation on this aspect in Appendix C.3, Table 3, where we vary the graph size, and the maximum in-degree / branching factor.
>
> These results show that CauKer performs best with moderate depth and branching, while very shallow or very dense graphs are less effective. We agree that this point is important, and in the revision, we will make the link to Appendix C.3 more prominent in the main text so that the role of the causal graph structure is clearer.
>
>
> ---
>
> Once again, we thank the reviewer for the thoughtful feedback and for the encouraging overall evaluation. We believe the clarifications and additional analyses described above will strengthen the final version of the paper (**highlight in orange**).

---

### Author Response · Authors · 2025-12-02

We would like to sincerely thank the AC for their efforts in handling the unexpected situation around the rebuttal process, and all reviewers for their thoughtful comments.

Reviewer **Ztoq** (score 8) was very positive about the work, highlighting the contribution and experimental design. In response to their questions, we clarified how CauKer handles multivariate structure conceptually, explained the extension to forecasting and added a detailed runtime breakdown showing that the SCM component introduces negligible computational compared to GP sampling.

Reviewer **zLKs** initially raised concerns mainly about the exact “zero-shot” protocol. We clarified the pretraining and evaluation pipeline, and added non-foundation baselines. After these clarifications and additions, their score was raised from 6 to 8.

With Reviewer **Mojr** (score 6), we had a very constructive exchange. In response to their questions about generator complexity, failure modes, and real-data scaling, we (i) described the design of the kernel/mean/activation banks, (ii) added new ablations where CauKer is forced to use only a single kernel family, (iii) discussed why UEA is not an ideal large-scale pretraining corpus and how CauKer can act as a scalable alternative, and (iv) added attention-rollout visualizations. The reviewer explicitly stated that these additions fully addressed their questions and “made the paper even stronger”.

Reviewer **uHPf** (score 4) focused on the specific challenges of moving from forecasting to classification. We clarified the methodological differences between forecasting-oriented generators and classification-oriented data (role of non-zero mean functions and SCM mixing for discriminative patterns), sharpened the positioning with respect to TabPFN-style synthetic pretraining. We also extended the evaluation to irregular multivariate clinical benchmarks and fine-tuning experiments on UCR. Following their follow-up comment on wording of P19 dataset evulation, we also replaced the phrase “slightly worse” with the exact number. While we have not received further comments from the reviewer, to the best of our understanding, all the raised issues have been addressed in our response.

Overall, our contribution is to demonstrate that a carefully designed causal-kernel synthetic generator (CAUKER) can pretrain classification TSFMs purely on synthetic data, achieving competitive zero-shot performance and clear data/model scaling laws while greatly reducing reliance on large real-world corpora.

We once again thank the AC for their careful consideration of our work and for their efforts in guiding the review process under these exceptional circumstances.

---

### Meta-Review · Area_Chair_nD9L · 2026-01-06

**Summary:**

Reviewers were broadly positive about the paper’s core idea—pretraining classification TSFMs purely on CAUKER synthetic data—and consistently highlighted the strength and breadth of the empirical evaluation, including clear data/model scaling laws and competitive zero-shot performance. Initial concerns focused on (i) novelty and positioning relative to prior synthetic-pretraining work (e.g., TabPFN-style SCM generators), (ii) clarity of the evaluation protocol, particularly the interpretation of “zero-shot”, (iii) generator complexity and reproducibility, and (iv) the scope of evaluation across tasks, datasets, and settings.

Through the rebuttal and revisions, the authors substantially clarified the evaluation protocol, added missing baselines and ablations, expanded experiments to multivariate and irregular settings as well as downstream fine-tuning, and sharpened the methodological positioning of CAUKER relative to prior work. As a result, the remaining concerns are largely about broader generality and future extensions rather than the validity of the reported findings. Overall, reviewers converged on the view that the paper presents a compelling and timely contribution to synthetic pretraining for time-series foundation models, supporting a positive recommendation.

**Reviewer Concerns:**

The rebuttal addressed many of the reviewers’ concerns by clarifying the zero-shot evaluation protocol, adding missing baselines, correcting wording and metric interpretation, and substantially expanding the empirical scope (including downstream fine-tuning, irregular/multivariate datasets, generator ablations, and runtime analysis).

**Reviewer Scores:**

see above

---

### Decision · Program_Chairs · 2026-01-26

Accept (Oral)